# Distinct tissue niches direct lung immunopathology via CCL18 and CCL21 in severe COVID-19

Ronja Mothes[1,2,21], Anna Pascual-Reguant[2,3,21], Ralf Koehler[2],
Juliane Liebeskind[2,3], Alina Liebheit[2,3], Sandy Bauherr[2], Lars Philipsen[4,5],
Carsten Dittmayer[1], Michael Laue[6], Regina von Manitius[1], Sefer Elezkurtaj[7],
Pawel Durek[8], Frederik Heinrich[8], Gitta A. Heinz[8], Gabriela M. Guerra[8],
Benedikt Obermayer[9], Jenny Meinhardt[1], Jana Ihlow[7], Josefine Radke[1,10,11,20],
Frank L. Heppner[1,12,13], Philipp Enghard[14], Helena Stockmann[14],
Tom Aschman[1], Julia Schneider[15], Victor M. Corman[15], Leif E. Sander[10,16],
Mir-Farzin Mashreghi[8], Thomas Conrad[17], Andreas C. Hocke[16],
Raluca A. Niesner[18,19], Helena Radbruch[1,22] & Anja E. Hauser[2,3,22] ✉

Prolonged lung pathology has been associated with COVID-19, yet the cellular and molecular mechanisms behind this chronic inflammatory disease are poorly understood. In this study, we combine advanced imaging and spatial transcriptomics to shed light on the local immune response in severe COVID-19. We show that activated adventitial niches are crucial micro-environments contributing to the orchestration of prolonged lung immuno-pathology. Up-regulation of the chemokines CCL21 and CCL18 associates to endothelial-to-mesenchymal transition and tissue fibrosis within these niches. CCL21 over-expression additionally links to the local accumulation of T cells expressing the cognate receptor CCR7. These T cells are imprinted with an exhausted phenotype and form lymphoid aggregates that can organize in ectopic lymphoid structures. Our work proposes immune-stromal interaction mechanisms promoting a self-sustained and non-resolving local immune response that extends beyond active viral infection and perpetuates tissue remodeling.

Adventitial niches are found in the outermost layer of intermediate-to-large blood vessels, as well as around airways, and represent specialized microenvironments where immune cells interact with stromal cells and the vasculature, to monitor the tissue status and modulate immune responses[1]. COVID-19 courses with severe histopathological changes and immune cell infiltration in the lung. These occur as a direct result of SARS-CoV-2 infection[2], but also indirectly through a dysregulated immune response[3], and lead to a functional breakdown of vasculo-epithelial barriers[4]. In later disease phases, tissue remodeling mechanisms aimed at repairing the micro-anatomical lung structure appear compromised in some individuals and lead to tissue scarring and lung fibrosis[5], which contribute to chronic respiratory symptoms and fibrotic disease[5,6]. An aberrant macrophage infiltration, along with increased numbers of fibroblasts in the lung has been proposed as a hallmark of severe COVID-19 disease[7,8]. In addition, alternatively activated macrophages accumulating in COVID-19 lungs show a pro-fibrotic signature[9], although the exact mechanisms how they promote fibrosis have yet to be clarified. Prolonged changes to the airway immune landscape with increased numbers of cytotoxic lymphocytes and B cells have recently been linked to sequelae of

COVID-19[10,11]. Thus, it is of utmost importance to shed light on the recruitment pathways that mediate excessive immune infiltration into the lungs and their implications for disease progression. Whether tissue pathology is the result of a viral reservoir in target organs, or caused by an autoimmune mechanism, or whether both phenomena contribute to the disease is a matter of ongoing discussion. Along with the significance of endothelial dysfunction in the pathomechanisms of COVID-19 disease[12,13], with recent reports highlighting post-acute, long-term cardiovascular manifestations[14] and chronic pulmonary vascular disease[6], the adventitial niche deserves special attention to elucidate the role of immune cells in sustained tissue damage.

Here, by combining histopathology, multiplex histology, electron microscopy (EM), three-dimensional light sheet microscopy (LSFM), single-nucleus RNA-sequencing (snRNASeq) and spatial transcriptomics (ST) we gained insights into the chemokine - chemokine receptor interactions occurring within lung niches during COVID-19. We propose a dysregulated immune-stromal crosstalk that promotes a self-sustained and non-resolving local immune response, extending beyond active viral infection. Our unique spatial approach identifies endothelial dysfunction as a trigger of tissue remodeling pathways within adventitial niches, which appear as seed points for fibrosis. Those niches emerge as specialized microenvironments that host T cells imprinted with exhausted profiles, as well as immune cell aggregates consisting of both B and T cells, in prolonged COVID-19.

## Results

### Severe COVID-19 is characterized by progressive lung fibrosis along with an accumulation of immune cells

In order to study the pathophysiological changes that occur in the lung associated to SARS-CoV-2 infection in our cohort, we combined several microscopy techniques and spatial transcriptomics (ST) in consecutive slides of post-mortem lung tissue samples obtained from COVID-19 donors. Lung samples from non-COVID-19-related pneumonia were included as controls. We stratified the COVID-19 cases into acute (1 to 15 days of disease duration), chronic (more than 15 days) and prolonged (7–15 weeks) (Table 1). We annotated intermediate-to-large vessels, airways, alveolar spaces, immune infiltrates and fibrotic areas in hematoxylin/eosin (HE)-stained lung sections (Supplementary Fig. 1a). These annotations served as landmarks for further spatial characterization of the local immune response within distinct functional lung microenvironments. COVID-19 lungs showed prominent alterations in the tissue composition and microanatomical structure, characterized by the accumulation of immune cell infiltrates and fibrotic areas (Fig. 1a and Supplementary Fig. 1a). These phenomena were particularly prominent in late disease stages, culminating in a lack of alveolar spaces. In line with that, immunofluorescence staining (IF) of adjacent tissue sections showed an accumulation of collagen deposits and of CD45+ immune cell aggregates in later disease phases (Fig. 1b; red arrows). ST analysis validated the exacerbated collagen deposition at the transcriptional level (Fig. 1c), and the acquisition of large image volumes by light sheet fluorescence microscopy (LSFM) further confirmed the increase in tissue density during prolonged COVID-19 disease (Fig. 1e, Supplementary Movie 1 and 2). The reticular ER-TR7 pattern formed a delicate, highly structured three-dimensional scaffold in the acute phase, while it appeared disorganized, dense and compact in the prolonged lungs.

We analyzed the same lungs by multi-epitope ligand cartography (MELC)[15,16], a multiplex microscopy technique that enabled us to apply a 44-parameter panel for immunofluorescence histology (Fig. 1d and Supplementary Fig. 1b). We pre-processed and segmented the multiplex microscopy immunofluorescence images to obtain quantitative information of the tissue composition at the single-cell level[17]. Total cell numbers and CD45+ cell counts increased significantly in the prolonged COVID-19 group (Fig. 1f, g). Notably, individuals in this group had resolved infection and lung samples were tested negative or

had only very low RNA load by qPCR, targeting the SARS-CoV-2 E gene. For active viral replication, subgenomic RNA (sgRNA) was used as a surrogate and we obtained negative results for all chronic and prolonged lung tissues analyzed. However, both chronic and prolonged cases showed aggravated lung damage and higher histopathological fibrosis scores based on HE/Elastica van Gieson (EvG) staining (Table 1), which is reflected by a positive correlation to the sample stratification. On the contrary, diagnosed bacterial pneumonia and acute respiratory distress syndrome (ARDS) neither correlated to the fibrosis score, nor to the disease group (Supplementary Fig. 1c), in line with a recent review reporting that post-COVID-19 interstitial lung disease develops independently of ARDS during the acute phase and can be understood as a new fibroinflammatory disease entity[18].

Thus, these data show that severe COVID-19 disease progression is characterized by lung fibrosis and that a prominent accumulation of immune cells occurs in prolonged cases, independent of active viral infection.

### Endothelial dysfunction in severe acute COVID-19 is an indirect effect of SARS-CoV-2 infection

We next aimed to further characterize the structural lung damage. In line with previous studies, we observed drastic lung vasculopathy in severe COVID-19 disease[4,13]. In our samples, the breakdown of the endothelial barrier was evident by the absence of CD31 staining in 5 out of 8 fields of view (FOVs) analyzed in the acute lungs, and showed an aberrant pattern in later disease phases (Fig. 2a). In line with previous reports, which point to a failure in remodeling of the alveolar epithelium after SARS-CoV-2 infection[2,4], we observed a lack of the prototypical monolayer structure in the pancytokeratin (PCK) staining (Fig. 2a). Indicative of lung fibrosis, the signals of smooth muscle actin (αSMA) and the reticular fibroblast marker ER-TR7 increased with disease duration (Fig. 2a). In addition, computational analysis of the multiplexed microscopy image data[17] (Supplementary Fig. 2a–c) revealed that the absolute numbers of endothelia, epithelia and fibroblasts were increased in the prolonged samples, pointing to restoration attempts, particularly within the endothelial compartment, where the differences were more pronounced (Fig. 2b).

Consistent with the loss of CD31 in acutely infected lungs, we observed several vessels with detachment of endothelial cells by electron microscopy (Fig. 2c) and a downregulation of several endothelial-related transcripts, including *PECAM1*, in the same disease group (Fig. 2d). The vasculopathy was also associated with a prominent coagulopathy and thrombotic events (Fig. 2e, Supplementary Movie 3). Despite all efforts, including ultrastructural (Supplementary Fig. 3d) and immunohistochemical analyses (Supplementary Fig. 3e) according to best practice standards[19], we could not find sufficient signs of pulmonary endothelial SARS-CoV-2 infection, explaining the extensive endothelial dysfunction observed. Thus, we conclude that endothelial barrier disruption, a prominent characteristic of severe acute COVID-19, is an indirect effect of SARS-CoV-2 infection, which is subject to restoration in later disease phases.

### Activated lung adventitial niches are fibrotic foci in COVID-19

To understand the mechanisms underlying the functional changes within the stromal, epithelial and vascular compartment, we performed gene set enrichment analysis (GSEA). Several pathways related to endothelial activation and remodeling, together with epithelial-to-mesenchymal transition, collagen biosynthesis and collagen modifying enzymes were induced upon COVID-19 in the lungs, particularly in the prolonged phase (Fig. 3a). Furthermore, dimensionality reduction and unsupervised clustering analysis of the ST data revealed 9 spot clusters that represented discrete lung microenvironments with distinct transcriptional signatures indicative of endothelia (C3 and C8), epithelia (C0 and C6), stroma (C2 and C5), or distinct immune lineages (C1: macrophages and C4: plasma cells) (Fig. 3b). The clusters were

**Table 1 | Clinical data table**

| Study ID | Age range | Sex | PMI [h] | Disease duration [days] | days at ICU | Invasive ventilation | ECMO | Catecholamines | Corticosteroids | ARDS | Bacterial pneumonia | SARS-CoV2 ante mortem | Mean log10 Sars-CoV-2 GE/10000 diploid cells (LUNG \| LN) | Subgenomic RNA (LUNG \| LN) | Nucleocapsid IHC (LUNG) | Prussian Blue (iron) lung stain | Fibrosis score (mean)/of N slides | COVID-19 as part of main cause of death after autopsy[a] | Technique |
|---|---|---|---|---|---|---|---|---|---|---|---|---|---|---|---|---|---|---|---|
| control case 1 | 90-94 | F | 70 | 1 | 1 | yes | no | no | no | no | yes | Negative | N/A \| N/A | N/A \| N/A | 0 | + | 2,83/2 | no | MELC (Lung&LN), ST |
| control case 2 | 55-59 | M | 36 | 55 | 7 | no | no | no | yes | no | yes | Negative | N/A \| N/A | N/A \| N/A | 0 | + | 1,9/1 | no | MELC (Lung), ST |
| control case 3 | 65-69 | M | 40 | 28 | 7 | yes | yes | yes | no | no | yes | Negative | N/A \| N/A | N/A \| N/A | 0 | ++ | 1,08/1 | no | ST |
| control case 4 | 75-79 | M | 50 | N/A | 0 | no | no | no | no | no | yes | N/A | N/A \| N/A | N/A \| N/A | N/A | N/A | N/A | no | snRNAseq |
| acute case 1 | 80-84 | M | 82 | 11 | N/A | no | no | no | no | yes | no | Positive | 6,67 \| 4,85 | yes \| yes | +++ | ++ | 2,93/5 | Ib | MELC (Lung), ST, snRNAseq |
| acute case 2 | 90-94 | M | 22 | 12 | 3 | no | no | no | yes | yes | yes | Positive | 5,25 \| 3,11 | yes \| no | ++ | ++ | 1,88/1 | Ic | MELC (Lung&LN), ST |
| acute case 3 | 75-79 | M | 30 | 13 | 5 | yes | no | yes | yes | no | yes | Positive | 5,89 \| 3,33 | yes \| yes | ++ | NA | 2,73/1 | Ib | MELC (Lung&LN), ST, snRNAseq, LSFM |
| chronic case 1 | 60-64 | M | 10 | 19 | 13 | yes | no | yes | yes | yes | no | Positive | 2,83 \| 1,33 | no \| no | + | ++ | 2,68/1 | Ic | MELC (Lung&LN), ST, snRNAseq |
| chronic case 2 | 65-69 | M | 30 | 34 | 20 | yes | yes | yes | no | yes | yes | Positive | 1,36 \| 2,08 | no \| no | 0 | + | 4,71/4 | Ic | MELC (Lung), ST |
| chronic case 3 | 55-59 | F | 38 | 27 | 20 | yes | yes | yes | no | yes | yes | Positive | 1,5 \| 2,02 | no \| no | 0 | 0 | 4,03/5 | Ic | MELC (Lung&LN) |
| chronic case 4 | 60-64 | M | 8 | 28 | 0 | no | no | no | no | no | yes | Positive | 2,77 \| 0 | no \| N/A | N/A | NA | 4,23/1 | Ib | ST |
| chronic case 5 | 65-69 | F | 17 | 34 | 30 | yes | yes | yes | yes | yes | yes | Positive | 0,42 \| 0 | no \| N/A | N/A | N/A | N/A | II | snRNAseq |
| prolonged case 1 | 70-74 | M | 116 | 51 | 42 | yes | no | yes | no | yes | no | Positive | 1,7 \| 1,37 | no \| no | 0 | +++ | 3,48/5 | Ic | MELC (Lung&LN) |
| prolonged case 2 | 75-79 | M | 53 | 51 | 51 | yes | no | yes | yes | yes | no | Positive | 0,58 \| 1,49 | no \| no | 0 | ++ | 3,07/5 | Ic | MELC (Lung) |
| prolonged case 3 | 75-79 | M | 84 | 107 | 105 | yes | no | yes | no | no | no | Positive | 0 \| 0 | N/A \| N/A | N/A | NA | 2,62/1 | Ic | MELC (Lung&LN), ST, snRNAseq |
| prolonged case 4 | 70-74 | M | 40 | 51 | 48 | yes | yes | yes | no | yes | yes | Positive | 1,9 \| 1,26 | no \| no | 0 | +++ | 5,58/5 | Ic | MELC (Lung), ST, LSFM |
| prolonged case 5 | 70-74 | M | 3 | 78 | 72 | yes | yes | yes | no | yes | yes | Positive | 1,31 \| 0 | no \| N/A | 0 | +++ | 5,41/1 | Ic | MELC (Lung&LN), ST, LSFM, snRNAseq |

[a]As main part of cause of death we regarded immediate causes of death (Ia), conditions leading to cause of death (Ib), underlying cause (Ic), and further relevant conditions (II) that may have contributed to fatal outcome in analogy to the World Health Organization (WHO) guidelines. Briefly, the immediate cause of death represented the condition (disease, injury or complication) that preceded death most directly. The condition leading to the cause of death indicated a sequence with an etiological or pathological basis that prepared the way for the immediate cause of death by damage to tissues or impairment of organ function. Underlying cause was defined as the earliest condition that started the sequence between health and death.

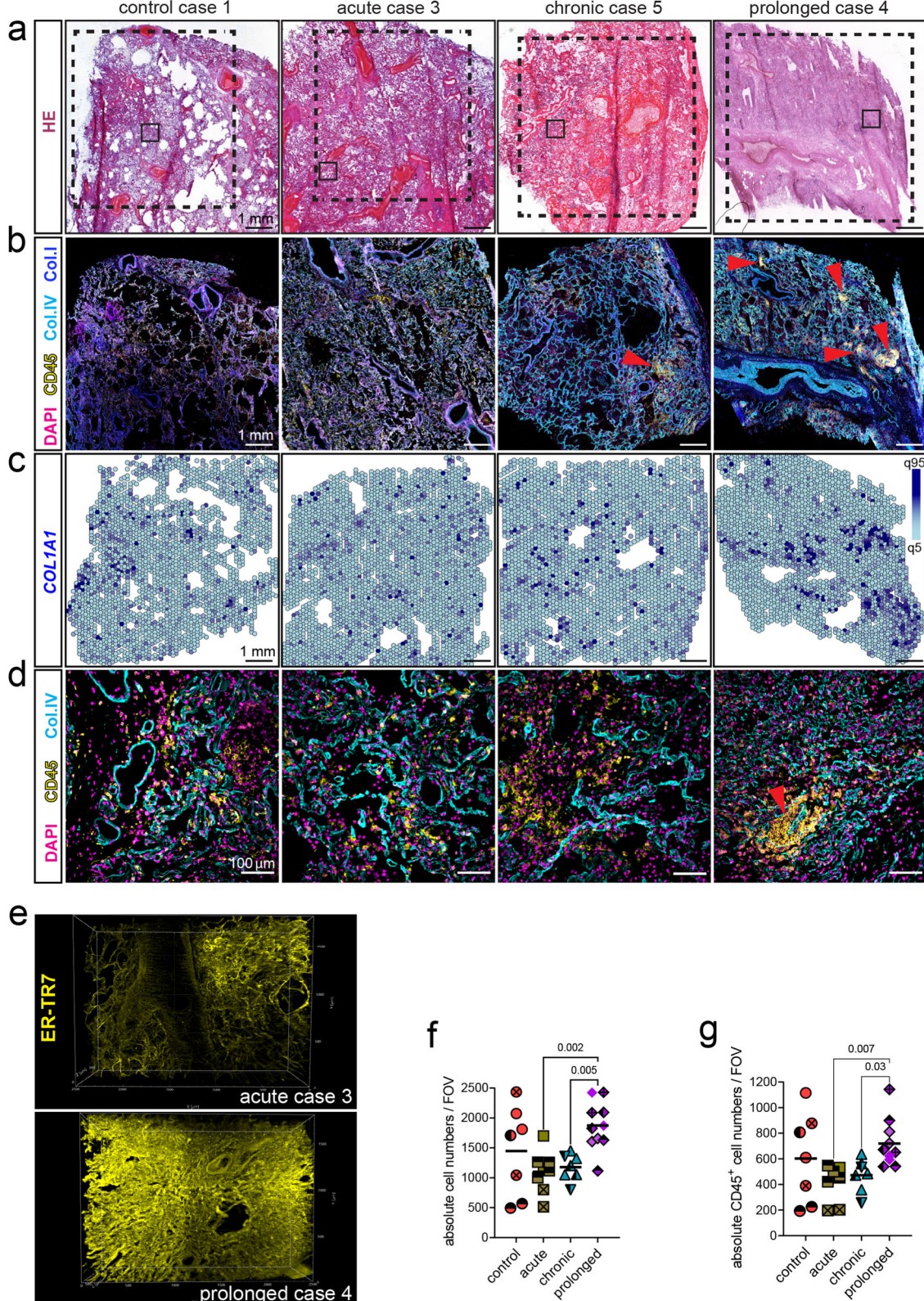

visualized in the UMAP space and the top 25 differentially expressed genes (DEG) were displayed on a heat map (Supplementary Fig. 3a–c). C5 was characterized by transcripts related to contractility and cytoskeleton rearrangements and was allocated to intermediate-to-large vessel landmarks and bronchi (Fig. 3c and Supplementary Fig. 3d), suggesting myofibroblast identity. The endothelial C8 was spatially distributed around those landmarks and adjacent to C5.

C8 showed high expression of *CCL21*, a chemokine that was hardly detectable in acute COVID-19 samples, but highly upregulated in later disease phases (Supplementary Fig. 3e). In addition, C8 was enriched with complement-related transcripts and *PTX3*, encoding for TNF-α induced protein 5 (Fig. 3b and Supplementary Fig. 3c). This protein promotes fibroblast differentiation[20] and is involved in complement activation, angiogenesis and tissue remodeling[21]. Indeed, these regions

**Fig. 1 | Severe COVID-19 is characterized by progressive lung fibrosis along with an accumulation of immune cells. a–d** Tissue sections of lung samples from a non-COVID-related pneumonia donor and COVID-19 donors at different time-points after disease onset were analyzed using spatially resolved techniques ($n = 12$ tissue sections). Each column represents a series of consecutive sections from the same donor. **a** HE staining shows tissue structure in whole sections. Dotted lines represent areas analyzed by confocal microscopy (**b**) and by spatial transcriptomics (ST; **c**). Small squares depict areas analyzed by multiplex microscopy (**d**). **b** Corresponding immunofluorescence (IF) images showing CD45 in yellow, DAPI in magenta, Collagen I (Col.I) in cyan and Collagen IV (Col.IV) in blue. **c** Relative expression of *COL1A1*, shown as Log2 fold change between the 5th and the 95th

quantile and displayed on the spots, as analyzed by ST. **d** Corresponding IF images showing DAPI in magenta, CD45 in yellow and Collagen IV in cyan. **e** Light sheet fluorescence microscopy (LSFM) acquisition of an acute and a prolonged COVID-19 lung samples stained with the fibroblast marker ER-TR7 (yellow) ($n = 4$ samples). See also Supplementary Movie 1 and 2. Dot plot depicting the absolute cell numbers (**f**) and CD45+ cell counts (**g**) per field of view (FOV), which represent 665 ×665 μm, analyzed by multiplex microscopy. Data (M ± SD) are analyzed by one-way ANOVA with Fisher´s LSD test, $F_{(3, 28)} = 5.09$, $p = 0.006$ (f), and $F_{(3, 28)} = 3.391$, $p = 0.03$ (g). Various filled-symbols represent distinct donors. Source data are provided as a Source Data file. (**d**, **f** and **g**) ($n = 32$ FOVs). See also Supplementary Figs. 1 and 2.

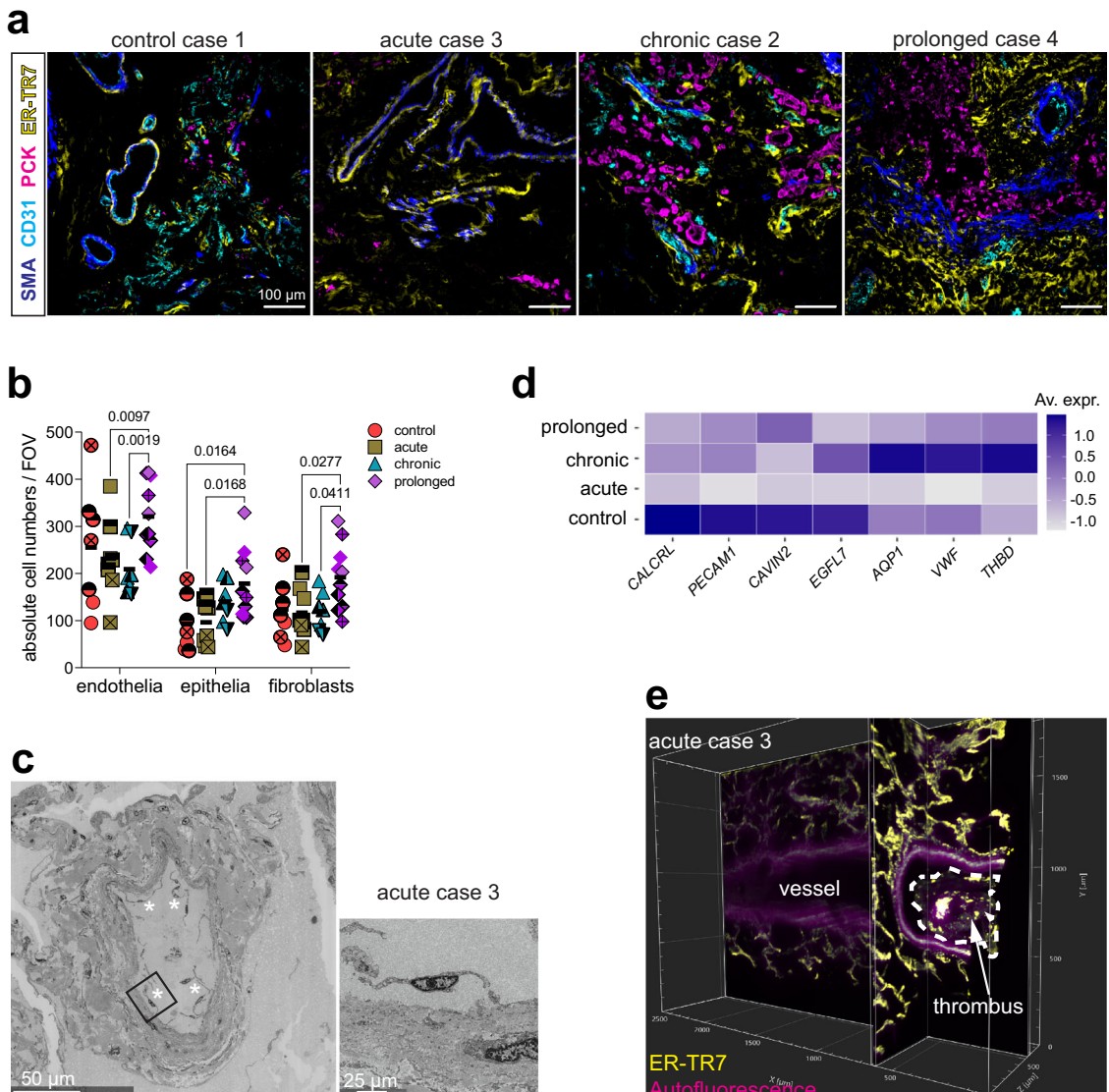

**Fig. 2 | Endothelial dysfunction in severe acute COVID-19 is an indirect effect of SARS-CoV-2 infection. a** Multiplex microscopy immunofluorescence (IF) images depicting smooth muscle actin (αSMA) in blue, CD31 in cyan, pancytokeratin (PCK) in magenta and ER-TR7 in yellow in one representative field of view (FOV) in lung tissue for each disease group. **b** Dot plot of the absolute cell numbers per FOV of endothelial cells, epithelial cells and fibroblasts in each disease group. Various filled-symbols represent distinct donors. Data (M ± SD) are analyzed by two-way ANOVA with Fisher's LSD test, $F_{(3, 84)} = 8.124$, $p < 0.0001$. Source data are provided as a Source Data file. (**a**, **b**) ($n = 32$ FOVs). See also Supplementary Figs. 1B and 2A–C. **c** Ultrastructural image of autopsy lung tissue from one acute donor shows a large

vessel with multiple detached endothelial cells within its lumen (white stars). Digitally magnified region of the boxed area shows a detaching endothelial cell. ($n = 6$ acute lung samples). **d** Heat map display of the average expression level of several endothelia-related transcripts for all samples in each disease group, as analyzed by spatial transcriptomics (ST) ($n = 12$ tissue sections). **e** Orthogonal slice of a light sheet fluorescence microscopy (LSFM) acquisition of an acute COVID-19 lung sample stained with ER-TR7 (yellow). Tissue autofluorescence (magenta) allows visualization of a large vessel with a macrothrombus (yellow and magenta) attached to the vessel wall. ($n = 4$ samples). See also Supplementary Movie 3.

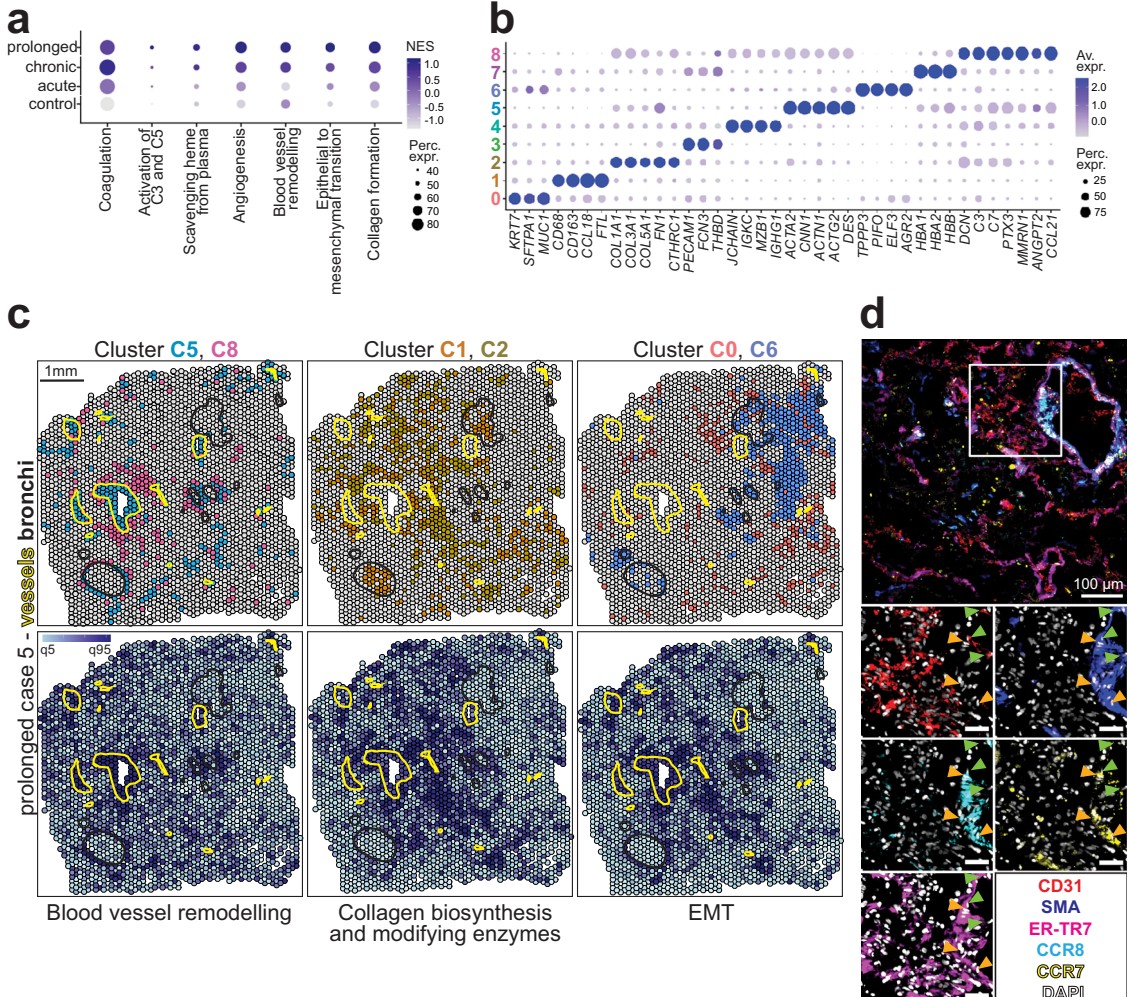

**Fig. 3 | Activated lung adventitial niches are fibrotic foci in severe COVID-19.**
**a** Dot plot depicting the normalized enrichment scores (NES) and the percentage of spots under tissue expressing relevant transcriptional pathways for each disease group, as analyzed by gene set enrichment analysis (GSEA) of spatial tran-scriptomics (ST) data. Coagulation, Angiogenesis and Epithelial to mesenchymal transition (Hallmark), Activation of C3 and C5, Scavenging of heme from plasma and Collagen biosynthesis and modifying enzymes (Reactome) and Blood vessel remodeling (GOBP) are shown (n = 12 tissue sections). **b** Dot plot depicting theNES and percentage of expression of several relevant transcripts among the top 25 differentially expressed genes (DEG) within each ST cluster, as shown in Supple-mentary Fig. 3A–C. **c** Color-coded tissue spots from one exemplary tissue section of a COVID-19 prolonged case analyzed by ST, depict the spatial distribution of

relevant ST clusters as shown in (**b**) and Supplementary Fig. 3a–c, and the NES from the Blood vessel remodeling pathway (GOBP), Collagen biosynthesis and modifying enzymes (Reactome) and Epithelial to mesenchymal transition (Hallmark). Anno-tations for intermediate-to-large vessels (yellow) and airways (black) are shown as outlines of these tissue landmarks. See also Supplementary Fig. 3d. **d** Multiplex microscopy immunofluorescence (IF) overlay depicting αSMA (blue), CCR8 (cyan), CCR7 (yellow), ER-TR7 (magenta) and CD31 (red) in a COVID-19 lung sample. The single channels for each extracellular staining together with DAPI (white) are shown for a region of interest, where scale bar = 40 μm (n = 8 FOVs). Green arrows indicate CD31$^+$SMA$^+$ER-TR7$^{+/-}$ cells and orange arrows, CCR8$^+$CCR7$^+$SMA$^+$ER-TR7$^+$ cells. (**a–c**) (n = 12 tissue sections).

of contractile structures (C5) and activated endothelia (C8) together spatially overlapped with areas in which the blood vessel remodeling pathway was enriched (Fig. 3c and Supplementary Fig. 3d). Hence, they hereafter will be referred to as activated adventitial niches. The mac-rophage cluster C1 was not spatially restricted to infiltrated bronchi, but also adjacent to these activated adventitial areas (Fig. 3c and Supplementary Fig. 3d). The macrophage transcriptional signature associated to C1 was accompanied by high *FTL* expression, along with *CD163* expression. The former is related to the storage of iron in a soluble state, the latter encodes for the heme-scavenger receptor (Fig. 3b) and has been recently linked to lung fibrosis in severe COVID-19[9]. We also detected an increase in Fe$^{3+}$ positive cells by Prussian blue staining along with disease duration (Table 1), indicative of leakage of plasma from disrupted vessels into the tissue. Besides its heme-scavenging signature, C1 was enriched in *CCL18* (Fig. 3b and Supple-mentary Fig. 3c), a chemokine known to be produced by alternatively

activated macrophages, inducing fibrosis via crosstalk with fibroblasts in idiopathic pulmonary fibrosis (IPF)[22]. Single nuclei RNA sequencing (snRNAseq) data obtained from some of the same samples and donors (Table 1)[23] confirmed that CD163$^+$ macrophages in COVID-19 are the cellular source of *CCL18* (Supplementary Fig. 3f). The macrophage population upregulated *CD163* in COVID-19 lungs compared to con-trols, and showed an enrichment of *CCL18* expression in later disease phases compared to the acute lungs. Linking CD163, CCL18 and tissue fibrosis, C2, which showed a clear fibrotic signature, displayed a spatial distribution similar to C1 within the lung parenchyma. Consistently, the collagen biosynthesis and modifying enzymes pathway overlapped with C1 and C2, in addition to the activated adventitial niches (Fig. 3c and Supplementary Fig. 3d). In contrast, epithelial C0 and C6 showed a very distinct localization pattern. We found micro-anatomical lung areas enriched in the epithelial-to-mesenchymal transition (EMT) pathway, which did not spatially overlap with the epithelial C0 and C6,

but rather with the adventitial C5 and C8. This suggests that tissue fibrosis not only arises from epithelial remodeling, but from the activated adventitial niches with a *CCL21* signature. Thus, it likely also represents endothelial-to-mesenchymal transition (EndMT). Several fibrosis-associated genes, such as *COL1A1*, *COL3A1* and *COL4A1*, the fibroblast-related transcripts *ACTA2*, *PDGFRA* and *CTHRC1* and the pro-fibrotic mediator *TGF-β* were upregulated in tissue of late COVID-19 disease phases (Supplementary Fig. 3e).

We next questioned whether the activated vasculature was actually able to respond to the CCL18 signal via the cognate receptor CCR8[24], as well as to CCL21 via CCR7. We indeed detected strong CCR8 and CCR7 expression in SMA[+] and ER-TR7[+] fibroblasts lining large vessels with disrupted endothelia (Fig. 3d, orange arrows). In addition, several cells of the vascular wall were positive for both endothelial and mesenchymal markers, CD31 and SMA and/or ER-TR7, another sign of EndMT (Fig. 3d, green arrows)[25–27]. However, while these stromal cells were the main subset expressing CCR8 in COVID-19 lungs, the CCR7 signal was not restricted to the stromal compartment. Instead, it was also observed in cells with a hematopoietic morphology (Fig. 3d).

Altogether, our results suggest that CCL21-enriched, activated adventitial niches with an EndMT signature function as seed points for tissue fibrosis, which is linked to the presence of CCL18[+] heme-scavenging macrophages.

### Activated lung adventitial niches host aggregates of exhausted T cells in prolonged COVID-19

Next, we aimed to further characterize the hematopoietic cells expressing CCR7 within COVID-19 lungs. We detected broad CCR7 and PD1 expression in large T cell clusters occurring within distinct adventitial niches in lung samples from the prolonged group (Fig. 4a, Supplementary Fig. 4a). Some of these cells co-expressed the proliferation marker Ki67 (Fig. 4a and Supplementary Fig. 4a; arrows). Such T-cell aggregates were found in 7 out of 12 FOVs of the prolonged lung samples analyzed by multiplex microscopy. Both CD4[+] and CD8[+] T cell counts were significantly enriched in the prolonged lungs, together with NK cells (Supplementary Fig. 2a–c and Fig. 4b). LSFM of COVID-19 lungs confirmed the increase in CD3[+] T cells with disease duration in the 3-dimensional space (Supplementary Movie 4 and 5).

Since CCL21 and its receptor CCR7 support T cell migration and homing to lymphoid tissues, but also the recruitment of activated T cells to the lung[28], we aimed to compare the local T cell response at the effector site to that of the paired draining lymph nodes (dLNs) (see Table 1 for paired organs analyzed; Fig. 4c, Supplementary Fig. 2a–c and Supplementary Fig. 4a–c). The CD4[+] T cell population from COVID-19 lungs showed higher expression of CD8 and cytotoxicity-related markers. The hallmark of both CD4[+] and CD8 + T cells in COVID-19 lungs was broad PD1 expression, together with CD16 (Fig. 4a, c and Supplementary Fig. 4a). While the former is regarded as a prototypical exhaustion marker, the latter is an antibody-dependent cellular cytotoxicity receptor. In contrast, both T cell populations in the paired COVID-19 dLNs showed classical activation markers, such as CD69, HLA-DR-DQ, ICOS and Ki67. The expression level of CD16 and PD1 increased with disease duration in both CD4[+] and CD8[+] T cell populations from the lungs, similar to CD57, another exhaustion marker. In parallel, the expression of activation markers in the dLN counterparts also increased (Fig. 4d). The upregulation of T cell activation in secondary lymphoid organs in later phases of COVID-19 mirrored the increase in CD141 and HLA-DR-DQ expression within the myeloid population of the dLN (Fig. 4e, f and Supplementary Fig. 4c, d). Chronic and prolonged COVID-19 dLNs showed abundant CD141[+]HLADR-DQ[+/-] dendritic cells (DCs) within the T cell areas, where multiple direct T – DC contacts could be observed (Fig. 4f, orange arrows). Notably, as in the lungs, the dLNs did not show any signs of active viral infection (Table 1), indicating that persistent T cell activation within dLNs occurred after the infection had been resolved.

Finally, and based on the occurrence of T cell aggregates within adventitial niches in the prolonged COVID-19 lungs, we compared the expression profile of T cells in distinct lung niches within those samples (Fig. 4g). We observed the highest expression of CD16, PD1 and the proliferation marker Ki67 in the CD4[+] T cell population located within the adventitial niche, in comparison to their epithelial niche counterparts. Co-expression of these markers was confirmed at the single-cell level within those areas, together with variable expression of activation and / or functional markers (Supplementary Fig. 4a). Similar characteristics have been recently described for precursor populations of exhausted T cells (T_PEX) in the context of chronic viral infections, where they are meant to balance viral clearance and immunopathology[29,30].

Thus, while T cell activation persists in the draining lymph nodes, CCL21-enriched lung adventitial niches in prolonged COVID-19 host T cell aggregates with an exhausted phenotype, suggesting local imprinting of the immune response.

### Adventitial niches aim at the formation of ectopic lymphoid structures in prolonged COVID-19 lungs

Some of the large immune cell aggregates observed in the prolonged lung samples appeared particularly densely packed, were associated to collagen[+] structures in the adventitial space and were sometimes directly attached to bronchi (Fig. 5a and Supplementary Fig. 5b). ST analysis revealed a unique transcriptional fingerprint within the area where CD45[+] cells were located (Fig. 5a, b). In line with our multiplex microscopy results, T_PEX-related transcripts such as *CD3D*, *TCF7* and *SELL* accumulated within that area. *CCL19*, *CCL21*, *LTB* and *CXCL13*, which are cytokines that play a crucial role in the organization of lymphoid structures[31] co-localized there. Along that line, the B cell master regulator *PAX5* strongly accumulated there, along with *CCR6*, which has been related to B cell activation and memory formation[32,33]. On the other hand, the immunoglobulin transcripts *IGHG1*, *IGHD*, *IGHA1* and *IGHM* were prominently expressed in airway-surrounding regions and lower levels of *IGHD*, *IGHA1* and *IGHM* were found within the lymphoid aggregate (Supplementary Fig. 5a). While *IGHA1* was robustly detected within all lung samples analyzed, *IGHA2* transcripts were mainly absent, in accordance with published data from blood[34] and lung[35]. *HLA-DRA* and *MKI67* transcripts were broader expressed across the tissue section, but also enriched within that region, indicating the occurrence of antigen presentation and proliferation (Supplementary Fig. 5a). Of note, macrophage-related transcripts, such as *MARCO* and *IL1B were* not expressed within the lymphocyte-enriched area, but rather within airway structures (Supplementary Fig. 5a). Similar large, dense and structured immune cell aggregates, showing the same unique transcriptional fingerprint reminiscent of ectopic lymphoid structures could be observed in 2 out of 3 prolonged lungs analyzed by ST (Supplementary Fig. 5b, c). In addition to that, smaller lymphoid aggregates containing both Pax5[+] B cells and CD4[+] T cells, in association to PNAd[+] high endothelial venules, were found in another prolonged case (Fig. 5c). Consistently, we also observed a significant increase in the absolute numbers of B cells and plasma cells in the prolonged COVID-19 group, compared to control tissues and to earlier disease stages (Fig. 5d). The occurrence of large lymphoid aggregates correlated with donor stratification and with the fibrosis score, but not with other clinical features, linking their emergence to disease progression and to the activated adventitial niches (Fig. 5e).

Altogether, our data suggest that activated adventitial niches are specialized microenvironments that support ectopic lymphoid structure formation in prolonged COVID-19 lungs.

## Discussion

The pathomechanisms behind prolonged COVID-19 are not well understood, although the consequences are challenging many individuals. In this study, we stratify the samples based on disease duration

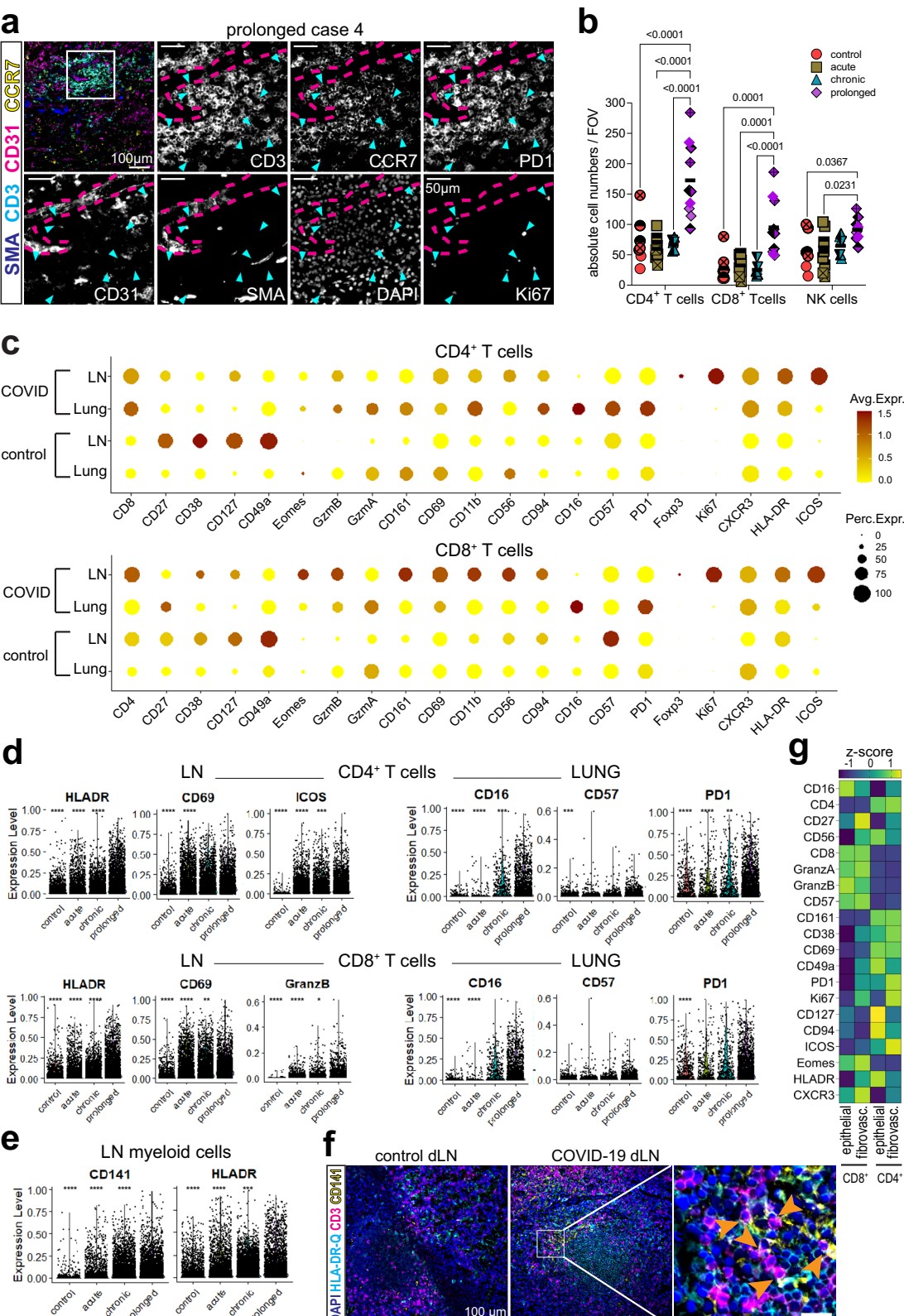

and include a prolonged group (7–15 weeks of disease duration post-infection), in which active viral infection could neither be reported in the lungs nor in the LNs. Yet, those individuals showed aggravated lung damage along with an increase in lymphocyte infiltration.

By combining advanced microscopy approaches coupled to single-cell computational analysis with ST techniques in autopsy tissues, we delineated the adventitial niche as a site, where a dysregulated

immune response leads to tissue remodeling in prolonged COVID-19. In line with the idea that endothelial dysfunction is a key pathological mechanism in COVID-19[12,13] and that endothelial cells are major participants in and regulators of inflammatory reactions[36], we propose endothelial-to-mesenchymal transition (EndMT) as a mechanism driving fibrosis. When activated endothelial cells undergo EndMT, they are transcriptionally reprogrammed, their tight cellular junctions are

**Fig. 4 | Activated lung adventitial niches host CCR7⁺ exhausted T cell aggregates in prolonged COVID-19. a** Immunofluorescence (IF) overlay depicts CCR7 (yellow), CD3 (cyan), αSMA (blue) and CD31 (magenta) in a prolonged COVID-19 lung. White square depicts region of interest (ROI), where enlarged single channels are shown, together with PD1, Ki67 and DAPI (*n* = 8 FOVs). See also Supplementary Fig. 4a. **b** Dot plot representation of absolute counts of CD4⁺ and CD8⁺ T cells and NK cells analyzed by multiplex microscopy. Data (M ± SD) are analyzed by two-way ANOVA with Fisher's LSD test, *F* (3, 84) = 29.56, *p* < 0.0001. See also Supplementary Fig. 2a–c, Supplementary Movie 4 and 5. **c** Dot plot showing protein expression of CD4⁺ and CD8⁺ T cells in lungs and lymph nodes of COVID-19 donors and controls. Color code indicates average mean fluorescence intensity (MFI) for each marker and dot size indicates percent expressed. **d** Violin plots show expression level of relevant markers in CD4⁺ and CD8⁺ T cells in lymph node and lungs for each disease

group. See also Supplementary Fig. 4A–C. **c–d** (*n* = 41 FOVs). **e** Violin plots show expression level of relevant markers in myeloid cells of lymph nodes for each disease group. See also Supplementary Fig. 4c. (*n* = 9 FOVs). **d–e** Data analyzed by one-way ANOVA with Wilcoxon test, comparing each group to the prolonged, where * *p* < 0.05, ** *p* < 0.01, *** *p* < 0.001 and **** *p* < 0.0001. **f** IF overlays depict DAPI (blue), HLA-DR-DQ (cyan), CD3 (magenta) and CD141 (yellow) in a control and a chronic lymph node. White square depicts the ROI, where orange arrows indicate direct cell contacts between T cells and CD141⁺HLA-DR-DQ⁺ dendritic cells. Scale bar: 20 µm (*n* = 9 FOVs). **g** Heat map display (z-score) of the MFI for each marker of CD4⁺ and CD8⁺ T cells within lung perivascular and epithelial niches in the prolonged group analyzed by multiplex microscopy. (*n* = 32 FOVs). Source data are provided as a Source Data file. See also Supplementary Figs. 2a–c and 4a.

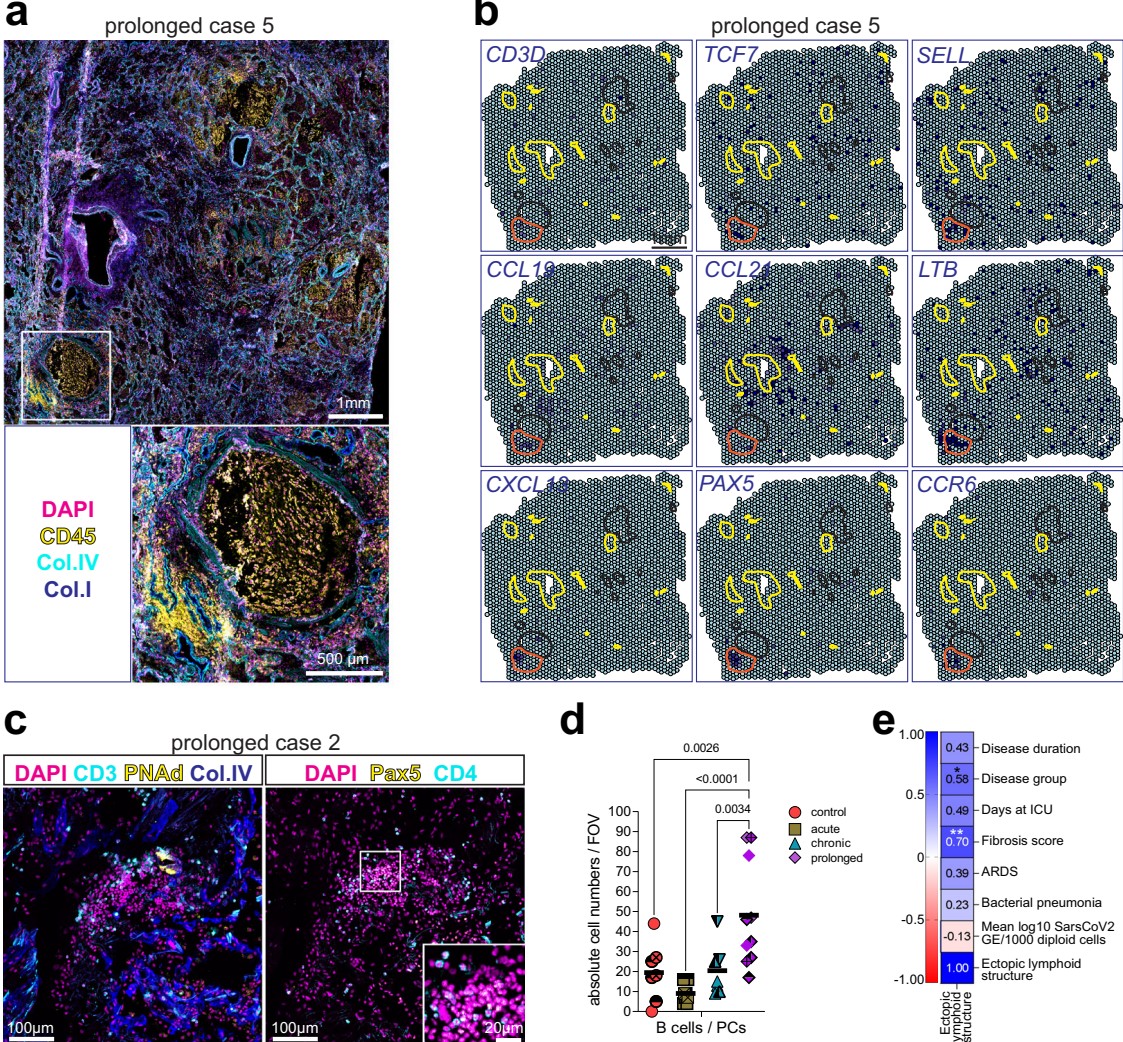

**Fig. 5 | Adventitial niches aim at the formation of ectopic lymphoid structures in prolonged COVID-19 lungs. a** Immunofluorescence (IF) image depicting CD45 (yellow), DAPI (magenta), Collagen IV (cyan) and Collagen I (blue). White square represents a region of interest (ROI), shown as an enlargement and depicting a highly infiltrated bronchus with a dense immune cell aggregate adjacent to it in the lower left corner. **b** The spatial distribution of ectopic lymphoid structure-related transcripts is shown in a color-coded fashion as normalized enrichment scores (NES) in overlay with relevant tissue landmarks: intermediate-to-large vessels (yellow line), airway structures (black line) and lymphoid structure (orange line). **a**, **b** (*n* = 2 prolonged lung samples). See also Supplementary Fig. 5. **c** IF images from the same area of a prolonged lung, where DAPI (blue), CD3 (red), PNAd (green) and

Collagen IV (Col. IV, white) are shown in the left panel, while DAPI (blue), Pax5 (magenta) and CD4 (cyan) are shown in the right (*n* = 9 FOVs). **d** Dot plot representation of the absolute counts of B cells / plasma cells per field of view analyzed by multiplex microscopy. Data (M ± SD) are analyzed by one-way ANOVA with Fisher's LSD test, *F* (3, 28) = 8.332, *p* = 0.0004 (n = 32 FOVs). Source data are provided as a Source Data file. See also Supplementary Fig 2a–c. **e** Heat map display of the correlation between relevant clinical parameters, donor stratification and fibrosis score with the presence of large immune aggregates. See also Supplementary Fig 1c. Pearson *r* values are shown in the plot with additional color code for positive (blue) or negative (red) correlations. Statistical significance (two-sided) is depicted with * for *p* = 0.039 and ** for *p* = 0.007.

disrupted and they turn into fibroblast-like cells. Consequently, they lose the expression of cell adhesion proteins, such as CD31, while mesenchyme-specific factors, including α-SMA, are upregulated[37]. All these features are evident in our samples (Figs. 2a, c and d, 3c, d, Supplementary Figs. 3c, d and 4a). In addition, we find that fibroblasts in COVID-19 lung adventitial niches express the chemokine receptor CCR8 (Fig. 3d). Its ligand, CCL18, has previously been linked to pulmonary inflammation and fibrosis[38] and it is the biomarker most consistently associated with negative outcomes in IPF[39]. CCL18 was previously reported to be enriched in SARS-CoV-2 RNA⁺ myeloid cells in the lungs of COVID-19-deceased individuals[2]. We have recently shown that alternatively activated, pro-fibrotic macrophages accumulate in fibrotic COVID-19 lungs[9]. In this study, we demonstrate that CD163⁺ macrophages accumulating in COVID-19 lungs produce CCL18 and spatially allocate this chemokine to the fibrotic patches (Fig. 3c and Supplementary Fig. 3d). Besides the known pro-fibrotic function of CCL18, this chemokine has recently been demonstrated to act as a cell-intrinsic factor promoting the longevity of tissue-resident macrophages in chronic inflammation[40]. Altered tissue biomechanics induced by fibrosis may further impact on immune cell location and function at those sites[41].

Our results also demonstrate an enrichment in *CCL21* in prolonged COVID-19 (Supplementary Fig. 3f), being spatially restricted to activated adventitial spaces with ongoing EndMT (Fig. 3c). High levels of CCL21 were found to be associated with persisting pulmonary impairment for at least 3 months after hospital admission with COVID-19[42]. CCR7, the receptor for CCL21, is expressed by IPF fibroblasts in contrast to normal lung fibroblasts, and CCR7⁺ IPF-derived fibroblasts respond to CCL21 with activation, migration, survival, and proliferation[43,44]. We show here that myofibroblasts in COVID-19 lungs also express CCR7 (Fig. 3d) and, thus, can respond to the elevated CCL21 signals within the adventitial niches, particularly in the prolonged disease phase. On the other hand, CCL21 is well known for its role in recruiting CCR7⁺ T cells to secondary lymphoid organs to organize the adaptive immune response[45,46]. CCL21 expression is also essential for the recruitment of effector T cells into the lungs[28]. In line with that, our results show broad expression of CCR7 within the T cell population in the lungs of prolonged COVID-19, the time when absolute numbers of both CD4⁺ and CD8⁺ T cells are found significantly increased compared to all other disease groups analyzed (Fig. 4a, b). In addition, the marked upregulation of CD16 in both lung CD4⁺ and CD8⁺ T cell populations and the enrichment of lung CD4⁺ cytotoxic T cells (Fig. 4c and Supplementary Fig. 4a) suggests that T cells acquire antibody-dependent cytotoxicity within the lung tissue, in line with published reports[47].

When comparing the T cell phenotype of the lung to that of the paired dLNs, our data point to a shift towards exhaustion within the inflammatory site, as opposed to a more conventional activated status in the secondary lymphoid organs (Fig. 4c, d and Supplementary Fig. 4a–c). In line with the persistent T cell activation profile, we also report an increase in CD141⁺HLA-DR-DQ⁺/· DCs in the LNs with disease progression, whereas these cells are undetectable in the lung (Fig. 4e, f and Supplementary Fig. 2a, c). CD141⁺ DCs represent a unique myeloid DC subset that cross-presents viral Ag after uptake of necrotic virus-infected cells[48]. The lack of evidence for viral persistence in both lungs and dLNs from the prolonged group suggests that ongoing cross-presentation of necrotic cellular antigens in secondary lymphoid organs causes the persistent T cell activation. On the other hand, the exhaustion shown by the lung T cell pool is characterized by broad and robust upregulation of PD1 in both, CD4⁺ and CD8⁺ T cell compartments, together with an increase in CD57 (Fig. 4a, c, d and Supplementary Fig. 4a). These immunomodulatory molecules are also expressed in proliferating cells, as shown by Ki67 positivity (Fig. 4a, and Supplementary

Fig 4a). Recent work has identified a link between exhaustion and the long-term maintenance of T cells[30], with an important role for TGF-β signaling in promoting the generation of $T_{PEX}$, a precursor population of exhausted T cells. Those cells are characterized by a restrained effector function and a metabolic profile that supports their persistence[49]. We indeed report an increase of TGF-β in the lungs of COVID-19 chronic and prolonged groups (Supplementary Fig. 3f), and previous studies have highlighted the essential role of TGF-β impacting on dysregulated immune functions in COVID-19[35,50]. Furthermore, CCR7 has recently been shown to mark a stem-like $T_{PEX}$ population localized in a CCL21-rich stromal environment[29]. The fact that CCL21 is highly enriched in regions of long-term T cell accumulation in the lungs further supports the idea that those areas represent $T_{PEX}$ niches. Thus, our findings extend the concept of $T_{PEX}$ niches to chronically inflamed human tissue and allocate the niches into adventitial spaces.

In prolonged cases, some of the areas enriched in the CCR7-ligands *CCL19* and *CCL21* showed the presence of *LTB* transcripts (Fig. 5b). Lymphotoxin beta (LT-β) has been shown to further enhance CCL21 expression upon airway challenge, which may represent a positive feedback loop to maintain the CCR7⁺ T cells[28] in those areas. The expression of the B cell attracting chemokine CXCL13[51] is enhanced in the same lung areas (Fig. 5b) and co-localizes with *PAX5*, along with evidence for a mucosal B cell profile, local B cell activation and antigen presentation. Together, the data presented here suggest the formation of ectopic lymphoid structures in prolonged COVID-19 lungs, and spatially link these structures to activated adventitial niches containing $T_{PEX}$ and showing signs of lung pathology. Whether there is also a functional link between T cell exhaustion and ectopic lymphoid structure formation will be a subject of future investigation.

Despite the non-negligible limitations of post-mortem studies, our results suggest that lung immunopathology in prolonged COVID-19 can occur within activated adventitial niches, independent of active viral infection. The mechanisms identified here are probably not unique to COVID-19 and might at least to some extent apply to other chronic diseases, for example, autoimmune conditions. We are aware that other mechanistic conclusions can be derived from these data, representing additional as well as alternative hypotheses. However, our work spatially links tissue fibrosis, T cell exhaustion and organization of ectopic lymphoid structures, which all represent hallmarks of chronic inflammatory diseases and cancer. It is appealing for us to further explore the extent of shared features between those various conditions at a spatial level, to assess whether those mechanisms may open up new roads for therapeutic strategies, beyond COVID-19.

## Methods
### Subject details
The lung and lung draining lymph node (dLN) samples included in this study have been collected in the Department of Pathology of Charité - Universitätsmedizin Berlin as part of the COVID-19 autopsy Biobank (see also Supplementary Table 1). This study was conducted in accordance with the declaration of Helsinki and with the approval of the Ethics Committee of the Charité (EA 1/144/927 13, EA2/066/20 and EA1/317/20) and the Charité - BIH COVID-19 research board. Autopsies were performed on the legal basis of §1 926 SRegG BE of the autopsy act of Berlin and §25(4) of the German Infection Protection Act. Autopsy consent was obtained from the families of the patients. Donor identities were encoded at the hospital before sharing for sample processing or data analysis. Some of the donors are part of the previously published cohorts in[9,19,52] and[53]. All COVID-19 donors showed a positive result in real-time reverse transcription polymerase chain reaction (RT-PCR) in oropharyngeal swabs at the time of hospital admission. All relevant characteristics and clinical information of the donors is presented in Table 1.

## SARS-CoV-2-specific PCR

For assessment of SARS-CoV-2 RNA of the lung and dLN samples, unfixed and, where possible, non-cryopreserved (i.e., native) tissue samples were used. RNA was purified from ~50 mg of homogenized tissue obtained from all samples by using the MagNAPure 96 system and the MagNAPure 96 DNA and Viral NA Large Volume kit (Roche) according to the manufacturer's instructions. The controls without COVID-19 were validated by Rhonda PCR rapid COVID-19 test (Spindiag) according to the manufactures protocol on ~50 mg of homogenized tissue together with a positive control.

Quantitative real-time PCR for SARS-CoV-2 was performed on RNA extracts with RT–qPCR targeting the SARS-CoV-2 E gene. Quantification of viral RNA was performed using photometrically quantified in vitro RNA transcripts[54]. Total DNA was measured in all extracts by using the Qubit dsDNA HS Assay kit (Thermo Fisher Scientific). The RT–qPCR analysis was replicated at least once for each sample.

As a correlate of active virus replication in the tested tissues, subgenomic RNA (sgRNA) was assessed by using oligonucleotides targeting the leader transcriptional regulatory sequence and a region within the sgRNA encoding the SARS-CoV-2 E gene[55]. All information is presented in Table 1.

## Standard histological and immunohistochemical techniques

Lung FFPE tissue blocks were taken at the day of autopsy and fixed for 24 h in 4% paraformaldehyde at room temperature. Routine histological staining (HE, EvG and Prussian blue) was performed according to standard procedures.

Immunohistochemical for nucleocapsid staining was performed 1:4000 with #HS-452 011 (Mouse; Synaptic systems) on a Benchmark XT autostainer (Ventana Medical Systems) with standard antigen retrieval methods (CC1 buffer, pH 8.0, Ventana Medical Systems) using 4-µm-thick FFPE tissue sections. Immunohistochemistry sections were evaluated by at least two board-certified (neuro-)pathologists with concurrence. To biologically validate the immunohistological stainings, control tissues harboring or lacking the expected antigen were used. Staining patterns were compared to expected results as specified in the Supplementary Fig. 2B, a semi-quantitative scoring was applied (0 = no positive cell, + = single positive cells, ++ = some positive cells, +++ abundant positive cells and debris) (see Table 1 and Supplementary Fig. 2B).

Fe3+ positive cells were scored using Prussian blue staining by two independent pathologists semi-quantitatively (0 = no positive cell; + = single cells; ++ = some cells and iron depositions; +++ = abundant positive cells and iron depositions).

Images were analyzed using a Zeiss Axiolab 5 microscope.

## Electron microscopy

Frozen autopsy lung samples were thawn, fixed with 2.5% glutaraldehyde and 2% formaldehyde in 0.1 M sodium cacodylate buffer and embedded in Epon according to a routine protocol including en bloc staining with uranyl acetate and tannic acid[52]. Large-scale digitization of ultrathin (70 nm) sections was performed using a Zeiss Gemini 300 field-emission scanning electron microscope in conjunction with a STEM detector via Atlas 5 software at a pixel size of 3–4 nm as previously described[56]. Regions of interest from the large-scale datasets were saved by annotation ('mapped') and then recorded at very high resolution using a pixel size of 0.5–1 nm. Other images of virus particles in autopsy lung tissue of the same patient were previously published[57–59]. We tried to optimize the sample handling for best preservation of the tissue, however effects mediated by autolysis and suboptimal fixation conditions cannot be excluded. We also analyzed two large-scale datasets of autopsy lung that we published previously[52] (see also http://www.nanotomy.org/OA/Krasemann2022eBioMedicine/index.html).

## Confocal microscopy

Cryosections of lung tissue were cut at 7 µm on a MH560 microtome, transferred onto Superfrost Plus Gold slides (Fisher Scientific, Jena, Germany) and air dried. Sections were surrounded with a PAP pen (Zymed Laboratories, Jena, Germany) and rehydrated with PBS for 10 min. After that, sections were blocked for 20 min at room temperature with 10% goat serum in PBS. For immunofluorescence staining, antibodies were diluted in staining buffer (PBS, 5% BSA and 0.01% Triton) and incubated at room temperature for 60 min. The following antibodies were used for staining: anti-Collagen I-PE, anti-Collagen IV-FITC, anti-CD45-Alexa Fluor 647 and DAPI. After staining, samples were washed three times with PBS for 5 min and mounted with ProLong gold mounting medium (Life Technologies, Waltham, MA). Images were acquired by confocal microscopy using a Zeiss LSM 880 microscope with × 20/0.5 NA (air) objective at room temperature.

## MELC

**Tissue preparation for MELC.** Fresh frozen lung and dLN tissue was cut in 5 µm sections with a MH560 cryotome (ThermoFisher, Waltham, Massachusetts, USA) on cover slides (24 × 60 mm; Menzel-Gläser, Braunschweig, Germany) that had been coated with 3-aminopropyltriethoxysilane (APES). Samples were fixed for 10 min at room temperature (RT) with freshly opened, electron microscopy grade 2% paraformaldehyde (methanol- and RNAse-free; Electron Microscopy Sciences, Hatfield, Philadelphia, USA). After washing, samples were permeabilized with 0.2% Triton X-100 in PBS for 10 min at room temperature and unspecific binding was blocked with 10% goat serum and 1% BSA in PBS for at least 20 min. Afterwards, a fluid chamber holding 100 µl of PBS was created using "press-to-seal" silicone sheets (Life technologies, Carlsbad, California, USA; 1.0 mm thickness) with a circular cut-out (10 mm diameter), which was attached to the cover slip surrounding the sample. Prior to each MELC experiment fresh washing solution consistent of PBS with 5% MACS BSA (Miltenyi Biotec) and 0.02% Triton X-100 was prepared. The sample was placed on the sample holder and fixed with adhesive tape followed by accurate positioning of the binning lens, the light path, as well as Köhler illumination of the microscope.

**MELC image acquisition.** MELC image acquisition was performed as previously shown[16,17]. In short, we generated the multiplexed histology data on a modified Toponome Image Cycler® MM3.2 (TIC) originally produced by MelTec GmbH & Co.KG Magdeburg, Germany[15]. The ImageCycler is a robotic microscopic system with 3 main components:(1) an inverted widefield (epi)fluorescence microscope Leica DM IRE2 (Leica Microsystems GmbH, Wetzlar, Germany), equipped with a Hamamatsu CMOS camera ORCA Flash 4.0 LT (Hamamatsu Photonics K.K., Hamamatsu City, Japan) and a motor-controlled XY-stage, (2) a pipetting system consisting of a XYZW-robot (Cybertron GmbH, Berlin, Germany) and a CAVRO XL3000 Pipette/Diluter (Tecan GmbH, Crailsheim, Germany), and (3) a TIC-Control software (MelTec) v3.0 for controlling microscope and pipetting system and for synchronized image acquisition.

Each MELC experiment is a sequence of iterative cycles, each consisting of four steps: (i) pipetting of the fluorescence-coupled antibody onto the sample, incubation and subsequent washing; (ii) cross-correlation auto-focusing based on phase contrast images, followed by acquisition of the fluorescence images 3-D stack (+/− 5 z-steps); (iii) photo-bleaching of the fluorophore; and (iv) a second auto-focusing step followed by acquisition of a post-bleaching fluorescence image 3D stack (+/− 5 z-steps). In each four-step cycle up to four fluorescence-labeled antibodies were used, combining PE, FITC, APC and DAPI and images were acquired for two fields of view (FOV) in order to maximize area and cell numbers analyzed for each experiment and sample.

**Antibody panel for MELC**. The antibodies used are listed in Supplementary Table 1. The antibodies were stained in the indicated order.

**MELC image pre-processing**. Image pre-processing was conducted as previously described[17]. In short, the reference phase-contrast image taken at the beginning of the measurement was used to align all images by cross-correlation. Afterwards, the signal of the bleaching image in each cycle and focal plane was used to subtract the background and correct the illumination of the fluorescence image obtained in the same cycle and focal plane[15]. Thereby, tissue auto-fluorescence and potential residual signal from the previous cycle were removed. In case of uncoupled antibodies and to account for unspecific signal of the secondary antibody used, we subtracted the fluorescence image of the secondary antibody stained and acquired before the corresponding primary antibody, instead of the bleaching image. Then, an "Extended Depth of Field" algorithm was applied on the 3D fluorescence stack in each cycle[60]. Images were then normalized in Fiji, where a rolling ball algorithm was used for background estimation, edges were removed (accounting for the maximum allowed shift during the autofocus procedure) and fluorescence intensities were scaled to the full intensity range (16 bit = $> 2^{16}$). The 2-D fluorescence images generated were subsequently segmented and analyzed.

**Cell segmentation and single-cell feature extraction**. Segmentation was performed in a two-step process as previously described[17], a signal-classification step using Ilastik 1.3.2[61] and an object-recognition step using CellProfiler 3.1.8[62]. Ilastik was used to classify pixels into three classes: nuclei, membrane, and extracellular matrix (ECM). A probability map for each class was generated. Classification of images regarding membranes and ECM was performed by summing up a combination of images, using markers expressed in the respective compartments, while only the DAPI signal was used to classify nuclei. The random forest algorithm (machine-learning, Ilastik) was trained by manual pixel-classification in a small region of one data-set (approx. 6% of the image) and applied to the rest without re-training. CellProfiler was subsequently used to segment the nuclei and membrane probability maps and to generate nuclei and cellular binary masks, respectively. These masks were superimposed on the individual fluorescence images acquired for each marker, in order to extract single-cell information (mean fluorescence intensity, MFI) of each marker per segmented cell and their spatial coordinates.

**MELC single-cell data transformation**. Data was transformed using the hyperbolic arcsine function with a scale argument of 0.2.

**Dimensionality reduction and clustering analysis**. After exclusion of cells expressing none of the markers included in the MELC panel by setting a cut-off at MFI = 0.15, protein expression of 39.074 single cells from 14 lung samples and 32.258 single cells from 7 paired lung dLN samples was analyzed. Seurat package 4.0.0 was used in R (R Core Team (2021, 2021) (https://www.R-project.org/)[63] to perform mean centering and scaling, followed by principal component analysis (PCA), and reduced the dimensions of the data to the top 11 principal components in the lung and 10 principal components in the lung dLNs. Uniform Manifold Approximation and Projection (UMAP) was initialized in this PCA space to visualize the data on reduced UMAP dimensions. The cells were clustered on PCA space using the Shared Nearest Neighbor (SNN) algorithm implemented as *FindNeighbors* and *FindClusters* with *n.epochs = 500* and default parameters (*res = 0.8*) for the lung data set. We obtained 26 clusters that we merged to get relevant populations for our analysis based on canonical lineage markers. We ended up with 8 cell clusters that were manually annotated based on cell-type-specific markers found to be differentially expressed. Protein expression of the main cell clusters was visualized using *UMMAPPlot, DotPlot* and *VlnPlot* functions from Seurat. 7 cell

clusters in the lung dLN data set were obtained and visualized using the same algorithms and functions, but with *res = 0.2*.

**Niche analysis**. We compared the phenotype of the CD4$^+$ and CD8$^+$ T cell clusters located in the perivascular niche to the ones located in the epithelial niche. We defined the niche as the area of a 32 pixel-radius (equivalent to 10 μm; the average size of a hematopoietic cell) around each endothelial or epithelial cell, similar to previous analysis[17]. We extracted the MFI of each T cell-related marker for the sum of all perivascular niches and epithelial niches and displayed the expression profile as heat map showing the z-score values.

**Spatial transcriptomics**
Visualization of gene expression in lung tissue was performed using 10× Visium spatial gene expression kit (10× Genomics) following manufacturers protocol. The four capture areas in a 10x Genomics Visium Gene Expression slide consist of 5000 spots with DNA oligos for mRNA capture that have a unique spatial barcode and a Unique Molecular Identifier (UMI). Each spot has 55 μm diameter and can therefore capture mRNA from 1 to 10 cells. Tissue sections from the same fresh frozen human lung samples used for MELC and snRNA-seq were analyzed by ST. Briefly, control ($n = 3$) and COVID-19 lung samples ($n = 9$) from donors categorized based on disease durations (acute $n = 3$, chronic $n = 3$, prolonged $n = 3$) were cut into 10 μm sections using a MH560 cryotome (ThermoFisher, Waltham, Massachusetts, USA), and mounted on 10X Visium slides, which were pre-cooled to −20 °C. The sections were fixed in pre-chilled methanol for 30 min, stained with CD45-AF647, CD31-AF594 and DAPI for 30 min and imaged using an LSM 880 confocal microscope (Zeiss). The sections were then permeabilised for 10 min and spatially tagged cDNA libraries were constructed using the 10x Genomics Visium Spatial Gene Expression 3' Library Construction V1 Kit. cDNA libraries were sequenced on an Illumina NextSeq 500/550 using 150 cycle high output kits with sequencing depth of ~5000 reads per spot. Frames around the capture area on the Visium slide were aligned manually and spots covering the tissue were manually selected based on the immunofluorescence staining, using Loupe Browser 5.1.0 software (10× Genomics). Sequencing data was mapped to GRCH38-2020-A reference transcriptome using the Space Ranger software (version 1.3.0, 10× genomics) to derive a feature spot-barcode expression matrix.

Space Ranger outputs (feature barcode matrices and image data) from 31.801 barcoded spatial spots from twelve 10x Visium capture areas were filtered and integrated in R (version 4.1.0) with Seurat (version 4.0.4). Spots with at least 250 detected genes and less than 13% mitochondrial content were combined from each library. For that, SCTransform normalization was performed[64], in order to harmonize the Pearson residuals from each RNAseq data set. We selected *nfeatures = 3000* for downstream integration. Then, principal component analysis (PCA) was performed on the matrix composed of spots and gene expression (UMI) counts. The dimensions of the data were thereby reduced to the top 50 principal components. Uniform Manifold Approximation and Projection (UMAP) was initialized in this PCA space to visualize the data on reduced UMAP dimensions.

The spots were clustered on PCA space using the Shared Nearest Neighbor (SNN) algorithm implemented as *FindNeighbors* and *FindClusters* with parameters *k = 30* and *res = 0.3*. The method returned 12 spot clusters, one of which was mainly enriched with mitochondrial genes. We removed this cluster from the data set and performed again PCA with the top 50 principal components, *FindNeighbors* and *FindClusters* with parameters *k = 30* and *res = 0.3*. The method returned 9 spot clusters that represented lung microanatomical areas with distinct transcriptional signatures and that were then visualized on the UMAP space.

The integrated data was also used to perform Gene Set Enrichment Analysis (GSEA) with the *fgsea* package (version 1.18.0)[65], which compared the Wilcoxon-ranked genes from each data set to the Hallmark, Reactome and Gene Ontology gene sets from MSigDB (v7.4). Only significantly enriched genes (adjusted $p < 0.05$) were taken into account to measure the Normalized Enrichment Score (NES). We then performed single sample gene set enrichment analysis (ssGSEA) with the *escape* package (version 1.2.0)[66], where the normalized enrichment scores can be visualized on the spatial feature plot in Seurat. NES range was clipped to the first and last five percentile to enhance visibility.

### Single-nucleus RNA-sequencing

**Single-cell isolation and library preparation.** Each lung tissue sample was minced and placed in digestion medium (500 U/ml Collagenase, I.5 U/ml Dispase and 1 U/ml DNAse) for 1 h at 37 °C. Cells were filtered through a 70 μm strainer and enzymatic reaction was stopped by cold RPMI with 10% fetal bovine serum and 1% L-Glutamine. Cells were washed with 50 ml cold RPMI with 10% fetal bovine serum and 1% L-Glutamine and red blood cells were lysed using Red blood cell lysis solution (MiltenyiBiotec). Finally, cells were filtered using a 40 μm FlowmiR Cell Strainer (Millipore) and re-suspended in PBS supplemented with 2% fetal bovine serum at the concentration of 10.000 cells/μl for scRNA-Seq. The single-cell capturing and downstream library constructions were performed using the Chromium Single Cell 3′V3.1 library preparation kit according to the manufacturer's protocol (10x Genomics). Full-length cDNA along with cell-barcode identifiers were PCR-amplified and sequencing libraries were prepared. The constructed libraries were either sequenced on the Nextseq 500 using 28 cycles for read 1.55 cycles for read 2, and 8 index cycles, or on the Novaseq 6000 S1 using 28 cycles for read 1, 64 cycles for read 2, and 8 index cycles, to a median depth of 36000 reads per cell.

**Single-cell RNA sequencing and analysis.** The Cell Ranger Software Suite (Version 3.1.0) was used to process raw sequencing data with the GRCh38 reference. Single-cell RNA sequencing data analysis was performed in R (version 3.6.1) with Seurat (version 3.1.1). Cells with at least 500 and less than 5000 detected genes and less than 10% mitochondrial content were combined from each library. Library depth (total number of UMIs) was regressed out when scaling data and libraries from different donors were integrated using *IntegrateData*. Cluster annotation was performed using the Human Lung Cell Atlas reference dataset[67] and Seurat's *TransferData* workflow as published elsewhere[23].

### Light sheet microscopy

**Whole-mount staining and optical clearing of human lung tissue.** About 3 mm PFA-fixed human lung cubic samples were processed for tissue permeabilization and decolorization based on a recently published approach[68]. All steps were performed permanently shaken on a tube rotator (MACSmix™, Miltenyi Biotec). We incubated the samples first in 5 mL 25% w/v Quadrol solution (N,N,N′,N′-Tetrakis-(2-hydroxypropyl)-ethylendiamin, Sigma-Aldrich) for 2 days at 37 °C, followed by 5 mL 25% w/v Quadrol and 10% w/v CHAPS solution ((3-[3-cholamidopropyl)dimethylammonio]-1-1propanesulfonate, Sigma-Aldrich) for 5 days at 37 °C[68]. After initial decolorization and permeabilization, we followed the SHIELD protocol for PFA-fixed human samples[69]. Therefore, samples were incubated in SHIELD-OFF solution at 4 °C for 3 days and then placed in SHIELD-ON Buffer and SHIELD-Epoxy Solution in a ratio of 1:1 at RT for 24 h. SHIELD preservation is then completed and we washed the samples for 1 day in PBS with three buffer changes at 4 °C. Next, we actively cleared the samples for 3 days using stochastic electrotransport (SmartClear Pro, LifeCanvas Technologies) and performed the same washing step as before.

For immunostaining, lung samples were incubated in blocking buffer (0.2% Triton X-100/10% DMSO/5% FCS/PBS) at 37 °C for 1 day and then in antibody incubation buffer (0,5% FCS/3% DMSO/0.2%

Tween-20/PBS) at 37 °C for 7 days containing the respective antibodies (as indicated for the respective samples). We used following antibodies and dilutions for staining: Sytox-AF488 (1:40.000, Thermo Fisher), ER-TR7-AF546 (1st 1:200/2nd 1:100, Santa Cruz Biotechnology) or near-infrared (NIR) CD3-iFluor790 (1:50, AAT Bioquest, Inc., Sunnyvale, CA, USA). After staining samples were washed first with washing buffer (0.2%Tween-20/PBS) for 1 day at 37 °C and with PBS at 4 °C with three buffer changes each. Then, immunolabeled samples were incubated for index matching first in 50% EasyIndex (LifeCanvas Technologies, refractive index –RI- 1,456) in ddH$_2$O for 1 day and second in 100% Easy Index until transparency was reached within 1-2 days.

**Sample mounting and light-sheet microscopy imaging.** We slightly glued the human lung samples on a plastic plate and immersed the sample in 100% Easy Index solution in a 5 × 5 × 45 mm quartz cuvette (msscientific Chromatographie-Handel GmbH). For image acquisition, we immersed the smaller cuvette containing the lung sample in the bigger quartz cuvette (LaVision Biotec), filled with fused silica matching liquid with a RI of 1,459 (Cargille Laboratories).To acquire light-sheet images we used an Ultramicroscope II (LaVision BioTec) coupled to an Olympus MVX10 zoom body providing a zoom ratio from 0.63×–6.3× and a 2× dipping objective (Olympus MVPLAPO2XC/0.5 NA [WD = 5.6 mm, corrected]) resulting in a total magnification ranging from 1.36×–13.56×. The Ultramicroscope II featuring a lateral resolution of ~4 μm was equipped with an Andor Neo sCMOS Camera with a pixel size of 6.5× 6.5 μm$^2$ and a LaVision BioTec Laser Module featuring the following filter sets: ex 488 nm, em 520/50 nm; ex 561 nm, em 620/60 nm; ex 785 nm, em 835/70 nm. We used the 488 nm ex for generation of autofluorescent signals and detection of nuclear Sytox-AF488 signal, the 561 nm ex for detection of ER-TR7-AF546 signal and the NIR 785 nm ex for detection of CD3-iF790. We imaged the longest wavelength first to avoid photobleaching during imaging and performed a linear z-adaption of the ex. 785 nm laser. We used a total optical zoom of 8,61× and a z-step of 5 μm for all acquisitions. Bigger tile scans were acquired using 10% overlap along longitudinal x-axis and y-axis of the human lung sample and the laser power was adjusted depending on the intensity of the signal (in order to not reach the saturation.

**Image processing.** We separately acquired 16-bit grayscale TIFF images for each channel by light sheet microscopy with the ImSpector software (LaVision Biotec). Tiff stacks were converted (ImarisConverterx64) into Imaris files (.ims) and stitched by Imaris Stitcher. We used Imaris (Bitplane) for 3-D and 2-D image visualization, snapshot creation and movie generation. We performed background subtraction in accordance with the diameter of the respective structures to eliminate unspecific background signals.

All instruments, reagents, software and algorithms used for this study are detailed in Supplementary Table 1.

### Statistics & Reproducibility

During the period of 03/2020 and 09/2020, all possible prolonged COVID-19 donors with an autopsy at Charité Universitätsmedizin Berlin, where COVID-19 diagnosis was given and respective tissue was available for analysis were included in this study. Randomly (age and sex matched) acute, chronic and control cases were selected among the cases with available tissue. All donors with a pulmonary tumor, hematologic stem cell therapy or autoimmune comorbidity were excluded. We used the same exclusion criteria for controls, but with negative PCR and immunohistochemistry for SARS-CoV-2 as additional inclusion criteria. No statistical method was used to predetermine sample size and additional randomization is not relevant to our study, as sample stratification was performed based on disease duration to interrogate differences based on that particular clinical parameter. We analyzed postmortal tissue samples from 17 human donors (4 control,

3 acute cases of COVID-19 disease, 5 cases of chronic COVID-19 disease and 5 cases of prolonged COVID-19 disease).

One analyzed control case had to be excluded due to postmortem PCR and immunohistochemical proof of a clinical unremarkable SARS-CoV-2 infection.

At least 2 different areas from each sample have been analyzed by multiplexed microscopy. Experiments have been performed in a minimum of 3 replicates per disease group. All attempts at replication were successful, except for two out of fourteen tissue sections analyzed by ST, which gave very low cDNA amounts, were nevertheless used for library prep, but did not pass the QC afterwards. These two tissue sections and the data extracted from them were not used for further analysis or for manuscript preparation. Evaluation of histological and immunohistochemical staining (tissue annotations and fibrosis scoring) was performed separately by at least two independent raters that were blinded for disease duration. Multiplexed histology, electron microscopy, light-sheet microscopy, snRNAseq and ST data acquisition, processing and analysis were not blinded.

Statistical analysis was performed with GraphPad Prism® 9.2.0 (Graph Pad Software, LA Jolla, CA, USA). The tests used and the exact $p$ values are described within each figure and figure legend. Significance was defined by $p < 0.05$.

### Reporting summary

Further information on research design is available in the Nature Portfolio Reporting Summary linked to this article.

## Data availability

De-identified human/patient spatial transcriptomics data have been deposited at Gene Expression Omnibus (https://www.ncbi.nlm.nih.gov/geo/), under record GSE190732 and is publicly available. SnRNA-Seq data have also been deposited at GEO (https://doi.org/10.1183/13993003.02725-2021) and are accessible under records GSM5958253, GSM5958254, GSM5958255, GSM5958256, GSM5958257, GSM5958258, GSM5958259 and GSM5958260.Multiplexed histology data, as well as feature matrices and spatial coordinates required to re-analyze the data reported in this study are publicly available in the Zenodo repository under DOI: 10.5281/zenodo.7447490 and all linked / related identifiers. Source data are provided with this paper.

## Code availability

This study did not use any original code.

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

## Acknowledgements

The authors are grateful to the patients and their relatives for consenting to autopsy and subsequent research. We thank Ralf Uecker and Gudrun Holland for technical support. This study was supported by funding from the Deutsche Forschungsgemeinschaft, HA5354/10-1, HA5354/12-1, SPP1937 (HA5354/8-2) and TRR130 P17 to A.E.H., TRR 130 C01 and SFB 1444 P14 (to A.E.H. and R.A.N.). R.A.N. was supported by DFG 1167/5-1. H.R. was supported by DFG (RA 2491/1-1 and CRC 130 TP17). A.C.H. was supported by Berlin University Alliance GC2 Global Health (Corona Virus Pre-Exploration Project), BMBF (RAPID and

Organo-Strat, alvBarriere-COVID-19) as well as DFG (SFB-TR 84, B6 / Z1a), by the Berlin Institute of Health (BIH), Charité 3R, and Charité-Zeiss MultiDim. S.E., F.H., H.R. and R.M. were supported by BMBF (Defeat Pandemics, Organo-Strat), F.H. was supported by DFG under Germany's Excellence Strategy EXC-2049-390688087, as well as SFB TRR 167 and HE 3130/6-1. B.O. was funded through the BIH Clinical Single Cell Bioinformatics Pipeline. Furthermore, we thank the Charité Foundation (Max Rubner Preis 2016) for financial support to C.D. M.-F.M was supported by grants from the Leibniz Association (Leibniz Collaborative Excellence, TargArt and ImpACt), the Berlin Institute of Health (BIH) with the Starting Grant-Multi-Omics Characterization of SARS-CoV-2 infection, Project 6, Identifying Immunological Targets in Covid-19, the state of Berlin and the European Regional Development Fund (ERDF 2014–2021, EFRE 1.8/11, Deutsches Rheuma-Forschungszentrum) and the German Federal Ministry of Education and Research (CONAN and TReAT).

## Author contributions

A.E.H., H.R., A.P.R. and R.M. conceptualized the study. J.M, J.R, H.R. and F.L.H. organized the Biobank. M.L., C.D., S.E., G.A.H., B.O., J.I., F.H., P.E., H.S., T.A., J.S., V.C., L.E.S., M-F.M., T.C., A.C.H., L.P., H.R, R.N. and A.E.H. provided resources and expertise. G.M.G, J.L., A.L., R.M. and A.P.R. performed experiments. R.v.M, F.H, P.D., R.K., J.L., A.L., R.M. and A.P.R. analyzed the data. R.M., A.P.R., H.R. and A.E.H. interpreted the results and wrote the manuscript. H.S., T.A., B.O., S.B., R.K., R.M., A.P.R., M-F.M, R.N., H.R. and A.E.H. reviewed the manuscript.

## Funding

## Competing interests

L.P. is affiliated with BioDecipher GmbH, a company which develops systems for multiplexed fluorescence microscopy. The remaining authors declare no competing interests.

## Additional information

[1]Department of Neuropathology, Charité–Universitätsmedizin Berlin, corporate member of Freie Universität Berlin and Humboldt-Universität zu Berlin, 10117 Berlin, Germany. [2]Immune Dynamics, Deutsches Rheuma-Forschungszentrum (DRFZ), a Leibniz Institute, Charitéplatz 1, 10117 Berlin, Germany. [3]Department of Rheumatology and Clinical Immunology, Charité - Universitätsmedizin Berlin, corporate member of Freie Universität Berlin and Humboldt-Universität zu Berlin, 10117 Berlin, Germany. [4]Institute of Molecular and Clinical Immunology, Medical Center, Otto-von-Guericke University Magdeburg, Magdeburg, Germany. [5]Multi-Parametric Bioimaging and Cytometry (MPBIC) platform, Medical Faculty, Otto-von-Guericke-University Magdeburg, Magdeburg, Germany. [6]Centre for Biological Threats and Special Pathogens (ZBS), Robert Koch Institute, Berlin, Germany. [7]Institute of Pathology, Charité–Universitätsmedizin Berlin, corporate member of Freie Universität Berlin and Humboldt-Universität zu Berlin, Berlin, Germany. [8]Therapeutic Gene Regulation, Deutsches Rheuma-Forschungszentrum (DRFZ), a Leibniz Institute, Berlin, Germany. [9]Core Unit Bioinformatics (CUBI), Berlin Institute of Health at Charité-Universitätsmedizin Berlin, Berlin, Germany. [10]Berlin Institute of Health (BIH), Berlin, Germany. [11]German Cancer Consortium (DKTK), Partner Site Berlin, CCCC (Campus Mitte), Berlin, Germany. [12]Cluster of Excellence, NeuroCure, Berlin, Germany. [13]German Center for Neurodegenerative Diseases (DZNE) Berlin, Berlin, Germany. [14]Department of Nephrology and Medical Intensive Care, Charité-Universitätsmedizin Berlin, corporate member of Freie Universität Berlin, Humboldt-Universität zu Berlin, 12203 Berlin, Germany. [15]Institute of Virology, Charité–Universitätsmedizin Berlin, corporate member of Freie Universität Berlin and Humboldt-Universität zu Berlin and German Centre for Infection Research, Berlin, Germany. [16]Department of Infectious Diseases, Respiratory Medicine and Critical Care, Charité-Universitätsmedizin Berlin and German Center for Lung Research (DZL), Berlin, Germany. [17]Genomics Technology Platform, Max Delbrück Center for Molecular Medicine in the Helmholtz Association (MDC), Berlin, Germany. [18]Dynamic and Functional in vivo Imaging, Veterinary Medicine, Freie Universität Berlin, Berlin, Germany. [19]Biophysical Analysis, Deutsches Rheuma-Forschungszentrum (DRFZ), a Leibniz Institute, Berlin, Germany. [20]Present address: Institut für Pathologie, Universitätsmedizin Greifswald, Greifswald, Germany. [21]These authors contributed equally: Ronja Mothes, Anna Pascual-Reguant. [22]These authors jointly supervised this work: Helena Radbruch, Anja E. Hauser. ✉e-mail: anja.hauser-hankeln@charite.de

