## [Peer Review File · Nature Communications]

Distinct tissue niches direct lung immunopathology via CCL18 and CCL21 in severe COVID-19Reviewer #2 (Remarks to the Author):

The study by Mothes represents an intense interrogation of lung tissues in 13 patients postmortem divided into Acute, Chronic and Prolonged disease. The studies combine histopathology, multiplex histology, electron microscopy (EM), three-dimensional light sheet microscopy (LSFM), single-nucleus RNA-sequencing (snSeq) and spatial transcriptomics, representing a technological tour-de force. The atlas of cell type and gene expression in these cases is extremely valuable and provides potential insights in the mechanisms and cellular cascades involved in the disease process as well as the potential role of cytokines in the development of fibrosis as well as the development of unusual cellular niches.

This paper has some issues that need to be addressed, particularly the conclusions regarding active processes that are gleaned from static observations in post-mortem tissues. AN example is: 'Indirect endothelial damage upon severe COVID-19 induces profound tissue

remodeling within fibrovascular niches." The title:

"Local CCL18 and CCL21 expand lung fibrovascular niches and recruit lymphocytes, leading to tertiary lymphoid structure formation in prolonged COVID-19" is also an overstatement, and there is no real evidence to support action and reaction. While the data may be suggestive of certain processes in play, the paper is written in a way that these are firm conclusions, rather than what the data may suggest,

The notion that these observations are relevant to PASC is uncertain, as these are all severe cases, resulting in death. Even in the prolonged group (7-15 weeks of infection), 4 of the 5 cases showed ARDS. This is not what we usually associate with Post Acute Sequelae of CoVID-19 and is more in line with an extended acute/chronic disease process.

The idea that Prussian Blue staining is evidence of CD163 macrophage scavenging is also an overstatement as the data shows iron accumulation with increase in CD163 macrophages. We really don't know what the cells are actually doing. The CD16 data should consider that expression may be on natural killer cells.

Reviewer #3 (Remarks to the Author):

Mothes et al study post-acute lung sequelae of COVID-19 and find an inflammatory cascade of events occurring along disease progression within fibrovascular niches.

Overall, I find the manuscript is well written, technical sound, and timely. It represents an important work contributing to our knowledge of the relationship of COVID-19 disease and fibrosis, highlighting the significance of the long covid burden for patients and the health system.

I really enjoyed reading the manuscript and have no major concerns. The authors found a great balance of the important biological question and the multi-disciplinary quantitative microscopy and computational pipelines.

Along those lines, I wonder whether the author would think that a boot strap approach (comparing smaller and the complete FOV) quantitatively would strengthen the findings. This of course represents only in the case global spatial cell organisation be helpful.

I also wonder whether the authors may want to discuss in the discussion the influence of the biomechanics of the healthy versus fibrotic tissue may influence the distribution of the immune cells.

Minor comments:

(1) Please standardise the scale bars across all Figures. For example, there are the scale bars in all panels, which is not necessary. Scale bars are missing in Figure 3. Scale bars have different thickness in Figure 4. The scale bars are not defined in the Figure captions.

(2) I would suggest to rephrase sentence "Altogether, our data demonstrate that CCL21 production in fibrovascular niches eventually contributes to the formation of tertiary lymphoid structures in prolonged COVID-19 lungs." to "Altogether, our data demonstrate that CCL21 production in fibrovascular niches may contribute to the formation of tertiary lymphoid structures in prolonged COVID-19 lungs."

(3) Please don't use "Here, ..." in the discussion section as it is reserved for the last paragraph of the introduction.

Reviewer #4 (Remarks to the Author):

This is a complex paper in which the authors propose a pathophysiologic model of the "progression" of severe Covid-19 infection on the basis of extensive and detailed analysis of the histopathologic correlates of chronic/prolonged COVID-19 infection, taking advantage of specimens collected in an autopsy biobank. Using multiple sophisticated high throughput multi-epitope staining and spatial transcriptomic analyses, the authors attempt to demonstrate is that the progression of COVID-19 infection from acute, to chronic, to prolonged (with fibrosis, as well as presumptive inflammatory correlates of "long COVID") is driven by endothelial injury, tissue remodeling emanating from an expanding fibrovascular niche with a "CCL21 signature", and subsequent formation of tertiary lymphoid structures in the diseased lung. The role of the fibroblastic niche and perivascular stromal cells in the orchestration of maintenance of tissue homeostasis and responses to injury have been the subject of considerable recent interest in a number of organs and disease states. In this manuscript the underlying hypothesis is that inflammatory/fibrotic progression results from misdirected/dysregulated responses within the FVN, leading to widespread tissue remodeling, particularly after immune clearance of the virus.

As the authors point out, severe acute COVID-19 infection often results in severe lung injury (ARDS), which is associated with diffuse alveolar damage, endothelial cell disruption, breakdown of vascular epithelial barriers, immune cell infiltration, and tissue remodeling. And regardless of the specific infection or trigger for such severe tissue injury, ARDS carries a high mortality (particularly in those managed with mechanical ventilation and its associated lung injury), and it is also common for those fortunate survivors to have significant residual fibrosis, often persisting for months or years, sometimes not resolving but also rarely progressive.

This begs the question of which aspects of the pathophysiology are normal responses to tissue injury, and which are "aberrant". In this study, all samples are from individuals who died (though not necessarily all specifically from COVID), and the nature of this imitation, as acknowledged by the authors, are inherent in a study of this nature, but do pose a constraint to the degree to which such findings might contribute to an understanding of prolonged post-COVID injury/inflammation/fibrosis Similarly the degree to which any of the proposed mechanisms might contribute to symptoms collectively referred to as "long COVID" is even more problematic. In addition, the assumption that any of their observations are unique to COVID-19 is based in large measure on their limited control population: 4 individuals who had apparent bacterial pneumonia, as well as a number of comorbidities, leading to their demise, 2 of which underwent invasive ventilation and 2 of which did not. Importantly none of them had

ARDS, and their exact cause of death is not specified. As such, one can only reasonably surmise that the findings presented represent observations concerning acute lung injury/ARDS in general (though in 3 of the 13 COVID cases ARDS was absent, including one in the acute group, one in the chronic, and one in the prolonged). These considerations aside, there are major concerns with the data which limit the degree to which the interpretations are convincing:

A) An important concern which permeates this manuscript is the manner of data presentation; it would behoove the authors to consider how they might present the data in such a way as to make the points they would like to make more clearly and persuasively, which would enhance the ability of readers to more readily judge the validity of their findings and interpretations. For example in Fig 1 multiple sections are shown from control specimens, as well as acute infection (14 days), chronic (34 days) and prolonged (51 days). H&E stains depict widespread abnormalities at low power, and a small "field of view" (FOV) has been selected as potentially informative (and presumably representative) by blinded pathologists. Though the specific patients whose samples are shown are not identified (which would have been helpful) one can clearly see (not surprisingly) that intensity of collagen staining correlates with duration of illness. There are other observations in the other panels, which might have been easier to appreciate with some carefully placed arrows, since depiction of the "juxtaposition" of different colors (which largely underpins their interpretations) is not always as obvious as the authors seem to believe. A critical issue is trying to understand the degree to which there is quantitative (or semi-quantitative) support for what are purported to be representative examples of their findings (though the interpretations are variably convincing by staring at which colors are next to which other colors (usually without a corresponding brightfield or other landmarks). Much must be taken on faith, particularly when staining patterns are marginally convincing.

B) In figure 2A for example, images from one case among the 4 groups show staining for smooth muscle actin, CD31, pancytokeratin, and ER-PR 7 (fibroblast marker). One has to surmise based upon the comments in the paper (and the legend) that the area at the bottom image from the prolonged case (which appears to show fibroblast staining interposed with faint CD31 staining) must be what the authors are calling a "fibrovascular niche", although the smooth muscle actin does not appear to overlap the fibroblastic staining. Fig 2B shows the expression levels of a number of endothelial genes PECAM 1, claudin 5, and von Willebrand's factor, in heat map fashion, of "average expression in each disease group" by spatial transcriptomics. If the authors more clearly delineated how representative these findings are, it might be more convincing that these do not simply reflect selectively presented data as well as what is to be in concluded from this. The point seems to be that endothelial markers are most down regulated in the acute case compared to the others (and though the single panel shown from the acute case seems to demonstrate complete absence of CD31 (in a single FOV), the text indicates that 5 out of 8 FOV's had no apparent CD31 staining (the panel shown has no cytokeratin staining either, and it seems vanishingly that both of the images are truly representative, since the case being shown (acute case 3) reported had no ARDS, and it is stated the COVID-19 was not part of the main cause of death. Figure 2D is another image of this case stained only for the fibroblast marker, shown only against autofluorescence, and the legend indicates that this is a large vessel with a thrombus attached to the vessel wall. This is not at all obvious, nor is the basis for making this determination. The supplemental video, while quite stunning, is no more informative. The interpretation appears little more than a guess, and one which is minimally supportive of their basic hypothesis at that. Also puzzling is the notation regarding Fig 2F, stating that "the image shows collagen 1 and CD45 both in white to depict tissue structure". Where is this white staining? The vessels are (manually) outlined in yellow, the fibrosis in green, and the airways in blue. Using spatial transcriptomics, genes associated with "blood vessel remodeling" are shown to be distributed differently in the acute case compared to the prolonged case, though the region which appears to have the most significant aggregation of local expression are in 3 foci in the field, only one of which is around a vessel. The largest aggregate concentration appears off to the bottom

right of the image from the prolonged case, where there is no evidence of a vessel. One would like to see more robust support for what appears to be authors' "gestalt", which is not very convincing. Is there more rigorous evidence for this (as the focus of the manuscript)? Unsupervised clustering of gene expression from single cell sequencing is stated to demonstrate that cluster 5 and cluster 9 are co-located. Though cluster 5 appears most prominent around the vessel in the field, though cluster 9 is most impressively aggregated at the bottom right, near no vessel (cluster 9 is stated to include genes associated with "endothelial identity", which is puzzling. It is stated that cluster 5 and cluster 9 overlap with fibrotic landmarks (?) with the perivascular markers nearby, and this is simply not convincing. It is worth noting that Fig S4 apparently shows the highest expressing genes by cluster, but is completely unreadable, even with enlargement. Would it not be more reasonable to show 12 different heat maps (or spreadsheets) with the top 25 genes, in readable format?

C) Figure 3 purports to show that fibrotic lung areas are enriched in CD163 and CCL 18, demonstrated in chronic case 5 and prolonged case 5, and neither of these interpretations are convincing based upon data shown. Based on the expression level scale to the right, both genes appear to actually be most downregulated around the presumptive fibrovascular niche in the prolonged case. CCL21 expression is shown to be up regulated in the region in question, and that seems convincing enough in the prolonged case 5. In figure B fibroblast proliferation appears to center around the structure in question as is to a lesser extent expression of collagen biosynthesis genes and EMT (stated to be "epithelial-mesenchymal" transition in the Legend, though the text is focused on endothelial-mesenchymal transition). Figure 3C focuses on area of interest from a chronic case seemingly showing smooth muscle actin, CCR8, CCR7, ER-TR 7, and CD31. It's not clear what is being emphasized or offered as an example, but to the naked eye little of interest seems apparent.

D) Figure 4 shows a field from a prolonged case #4, stated to represent a FVN, in which the large panel depicts staining for smooth muscle actin, CD3, CD31, and CCR7, in the region focused upon for the enlarged subpanels shows only the patch of CD3 staining. The panels to the right are stated to show the 4 stains individually, though the larger panel clearly shows much higher expression of CD31 and SMA outside of the selected region. The figure legend is entitled "Activated fibroblastic niches of prolonged COVID-19 lungs host CCR7 positive T-cell aggregates and imprint them with an exhausted, T follicular helper-like phenotype", though most of the data shown are expression data of CD4 cells and CD8 cells in lungs versus lymph nodes. Only figure 4G shows specific expression of CD8 and CD4 cell expression levels in "fibrovascular" (defined as in 4A?) versus epithelial regions, though it does not appear that any conclusion can be drawn from these heat maps, at least as presented. Neither is it evident (or stated) that statistical analysis has been applied to any of Fig 4 other than 4B.

E) Fig 5 is a bit confusing since there are 2 figures labelled A and B, followed by 3 figures subsequently labeled A, B, and C. The legend for Fig 5 is entitled "CCL21+ fibrovascular niches aim at the formation of tertiary lymphoid structures in prolonged COVID-19 lungs". Sections shown are from prolonged case 5, a case noted to have been complicated by bacterial pneumonia (as noted in 2 of the 5 prolonged, and 4 of the 5 chronic cases). In Fig 5A(#1) CD163 and CCL 18 expression by lung macrophages is shown, and there is significant divergence in CCL18 in chronic and prolonged (it would be helpful to know whether these were from 2 regions of interest, different patients, or what exactly is being shown. In Fig 5A(#2) a region of interest is outlined which is an airway stated to be "a highly infiltrated bronchus" with a dense immune cell aggregate adjacent to it in the left lower corner. The bronchus presumably contains a mucous plug (possibly with CD45+ PMN?; the patient is listed as having had secondary bacterial pneumonia). And while the CD45+ cell aggregate near it may represent bronchus-associated lymphoid tissue, supported by the expression of CD3, PAX5, lymphotoxin-beta, CXCL13), it cannot be determined to be a "tertiary lymphoid structure" without architectural information. It is clearly a small lymphoid aggregate, based upon gene expression, it would seem entirely possible that the aggregate is forming in response to

the secondary infection. Surely there are other examples of "true tertiary lymphoid structures" if they are real (and there were 3 prolonged cases without secondary bacterial pneumonia). One might speculate that tertiary lymphoid structure may be in the process of forming in this panel, ("aiming"?) but surely some must have formed somewhere in one of these lung specimens.

F) Fig S6A shows multiple panels each with multiple overlapping stains, and it is quite difficult to visualize the overlaps that are interpreted to reflect "non-conventional" T cells. In the bottom right subpanel of prolonged case 4, it would appear that ICOS and CD161 are co-localized, but the total number of stained cells (at the resolution in the figure) is not enough to be convincingly interpretable. Do the authors have better, or more extensive, representative examples? Would DAPI staining help delineate the cells in question? Are there other data to support the conclusion?

G) The discussion is littered with statements that reflect considerable overinterpretation of morphometric data, but also a curious conflation of post-Covid lung injury/fibrosis with "long-Covid", and there is little to support a relationship between these descriptive entities. The text refers to co-expression of PD1 and ICOS, which would not be surprising, but there are no such data apparent. The "data strongly suggesting the formation of iBALT in prolonged Covid-19 lungs"..."linking tertiary lymphoid structures to lung pathology and fibrosis" is hardly justified by the data.

NCOMMS-22-09465A

Point-to-point-response to reviewers

Dear reviewers,

Thank you for taking the time to review our manuscript and for helping us to improve the quality by giving us helpful feedback and advice. We have carefully addressed the issues raised. Large portions of the text, including the title of the manuscript, were changed in order to address those issues, and also to broaden the perspective about the relevance of the pathomechanisms discovered here. We also decided to replace the term “fibrovascular” by “adventitial” throughout the manuscript (see response to reviewer 4).

We hope that you are satisfied with the changes we made to the manuscript.

As specified by the submission guidelines, you will find the figure legends for supplementary files (movies) and source data description here in this letter.

Please find the point-to point response below.

Sincerely,

Anja Hauser

Legends for Supplementary files

Supplementary Movie 1. LSFM of COVID-19 lung from acute case 3 stained with antibodies against ER-TR7 for visualization of the fibroblast and reticular fiber network.

Supplementary Movie 2. LSFM of COVID-19 lung from prolonged case 5 stained with anti-ER-TR7 for visualization of the fibroblast and reticular fiber network.

Supplementary Movie 3. LSFM of COVID-19 lung from acute case 3 showing a thrombus attached to the vessel wall. Yellow color represents ER-TR7 staining, signals shown in magenta result from autofluorescence.

Supplementary Movie 4. LSFM of COVID-19 lung from acute case 3 stained with sytox green (nuclei) in magenta and anti-CD3 in cyan.

Supplementary Movie 5. LSFM of COVID-19 lung from prolonged case 5 stained with sytox green (nuclei) in magenta and anti-CD3 in cyan.

Source data description

All source data concerning cell quantification from multiplexed imaging, as well as the data used to generate the correlation matrix and the top 25 DEGs have been uploaded as supplementary information

Reviewer #2 (Remarks to the Author):

The study by Mothes represents an intense interrogation of lung tissues in 13 patients postmortem divided into Acute, Chronic and Prolonged disease. The studies combine histopathology, multiplex histology, electron microscopy (EM), three-dimensional light sheet microscopy (LSFM), single-nucleus RNA-sequencing (snSeq) and spatial transcriptomics, representing a technological tour-de force. The atlas of cell type and gene expression in these cases is extremely valuable and provides potential insights in the mechanisms and cellular cascades involved in the disease process as well as the potential role of cytokines in the development of fibrosis as well as the development of unusual cellular niches.

We would like to thank the reviewer for this positive assessment of our work.

This paper has some issues that need to be addressed, particularly the conclusions regarding active processes that are gleaned from static observations in post-mortem tissues. An example is: 'Indirect endothelial damage upon severe COVID-19 induces profound tissue remodeling within fibrovascular niches.' The title: "Local CCL18 and CCL21 expand lung fibrovascular niches and recruit lymphocytes, leading to tertiary lymphoid structure formation in prolonged COVID-19" is also an overstatement, and there is no real evidence to support action and reaction. While the data may be suggestive of certain processes in play, the paper is written in a way that these are firm conclusions, rather than what the data may suggest.

We agree that the previous version of the manuscript contained some phrasing which could be interpreted as overstatement. We have changed the wording in the relevant places and also adjusted the title, to clearly delineate the results from interpreted causality.

The notion that these observations are relevant to PASC is uncertain, as these are all severe cases, resulting in death. Even in the prolonged group (7-15 weeks of infection), 4 of the 5 cases showed ARDS. This is not what we usually associate with Post Acute Sequelae of CoVID-19 and is more in line with an extended acute/chronic disease process.

The reviewer raises an important point. To address this, we have decided to not use the term PASC or Long-COVID-19 and to take the emphasis out of possible link between our results and the mechanisms behind PASC. In the results, we show that lung fibrosis and immune cell aggregates are prominent characteristics of prolonged severe COVID-19, and we show that ARDS do not correlate with our sample stratification. Worth to mention is that ARDS in the individuals from the prolonged group could be diagnosed in the acute phases and/or upon admission and not at time of death, though, when individuals were stratified as prolonged. Additionally, we cite at that point in the text a recent review reporting such fibroinflammatory lung conditions after COVID-19 disease, referred to as post-COVID-19 interstitial lung disease (PC-ILD), which develops independently of ARDS in the acute phase¹. In the discussion, we introduce our concept that the prolonged severe cases show, as the reviewer suggests, a chronification of inflammatory processes, which may occur mechanistically similar (albeit to varying extent) in various settings/severities following COVID infection and apply to other chronic diseases.

The idea that Prussian Blue staining is evidence of CD163 macrophage scavenging is also an overstatement as the data shows iron accumulation with increase in CD163 macrophages. We really don't know what the cells are actually doing.

This was worded in a misleading way in the manuscript. As the reviewer correctly points out, Prussian Blue indicates the accumulation of iron in the cells. Together with our finding that heme scavenging pathways are enriched in later disease phases, particularly in the regions around the contractile structures/vessels (evidenced by GSEA analysis of the ST data, Fig. 3A), this indicates the leakage of heme into the tissue and the consequent scavenging of the heme. In the new version of the manuscript, we separated our conclusions on scavenging evidenced by GSEA (which is considered as a functional indicator) and the Prussian blue observation, to avoid confusion. Further supporting this concept, we added now the Ferritin light chain (FTL) together with CD163, as two of the top 25 DEGs in C1, enriched with other macrophage-related transcripts and found around adventitial niches in the lung parenchyma and spatially associated to the fibrotic patches defined by C2 (Fig. 3C).

The CD16 data should consider that expression may be on natural killer cells.

We thank the reviewer for raising this important point. We are aware that NK cells express CD16, but our phenotyping in Fig. 4 is performed from the MELC clustering analysis, in which the T cell cluster was CD3 positive and separate from the NK cluster (CD3 negative, see Fig. S2a-c).

Reviewer #3 (Remarks to the Author):

Mothes et al study post-acute lung sequelae of COVID-19 and find an inflammatory cascade of events occurring along disease progression within fibrovascular niches.

Overall, I find the manuscript is well written, technical sound, and timely. It represents an important work contributing to our knowledge of the relationship of COVID-19 disease and fibrosis, highlighting the significance of the long covid burden for patients and the health system.

I really enjoyed reading the manuscript and have no major concerns. The authors found a great balance of the important biological question and the multi-disciplinary quantitative microscopy and computational pipelines.

We would like to thank the reviewer for his positive evaluation - we are very pleased.

Along those lines, I wonder whether the author would think that a boot strap approach (comparing smaller and the complete FOV) quantitatively would strengthen the findings. This of course represents only in the case global spatial cell organization be helpful.

We thank the reviewer for his great suggestion helping us to strengthen the statistical power of our data. Following his advice, we now show every single FOV as a data point in all graphs (See Fig. 1D, 2B, 4B and 5D).

I also wonder whether the authors may want to discuss in the discussion the influence of the biomechanics of the healthy versus fibrotic tissue may influence the distribution of the immune cells.

We thank the reviewer for this nice idea and have added a reference to a recent review discussing the impact of mechanical stimuli in the tissue on the immune cell subsets our work focuses on.

Minor comments:

(1) Please standardise the scale bars across all Figures. For example, there are the scale bars in all panels, which is not necessary. Scale bars are missing in Figure 3. Scale bars have different thickness in Figure 4. The scale bars are not defined in the Figure captions.

We homogenized scale bar appearance and added them, if they were missing.

(2) I would suggest to rephrase sentence "Altogether, our data demonstrate that CCL21 production in fibrovascular niches eventually contributes to the formation of tertiary lymphoid structures in prolonged COVID-19 lungs." to "Altogether, our data demonstrate that CCL21 production in fibrovascular niches may contribute to the formation of tertiary lymphoid structures in prolonged COVID-19 lungs."

During the revision of the text we decided to completely change this paragraph, so we removed that sentence.

(3) Please don't use "Here, ..." in the discussion section as it is reserved for the last paragraph of the introduction.

We have replaced the beginning of the sentence accordingly.

Reviewer #4 (Remarks to the Author):

This is a complex paper in which the authors propose a pathophysiologic model of the “progression” of severe Covid-19 infection on the basis of extensive and detailed analysis of the histopathologic correlates of chronic/prolonged COVID-19 infection, taking advantage of specimens collected in an autopsy biobank. Using multiple sophisticated high throughput multi-epitope staining and spatial transcriptomic analyses, the authors attempt to demonstrate is that the progression of COVID-19 infection from acute, to chronic, to prolonged (with fibrosis, as well as presumptive inflammatory correlates of “long COVID”) is driven by endothelial injury, tissue remodeling emanating from an expanding fibrovascular niche with a “CCL21 signature”, and subsequent formation of tertiary lymphoid structures in the diseased lung. The role of the fibroblastic niche and perivascular stromal cells in the orchestration of maintenance of tissue homeostasis and responses to injury have been the subject of considerable recent interest in a number of organs and disease states. In this manuscript the underlying hypothesis is that inflammatory/fibrotic progression results from misdirected/dysregulated responses within the FVN, leading to widespread tissue remodeling, particularly after immune clearance of the virus.

As the authors point out, severe acute COVID-19 infection often results in severe lung injury (ARDS), which is associated with diffuse alveolar damage, endothelial cell disruption, breakdown of vascular epithelial barriers, immune cell infiltration, and tissue remodeling. And regardless of the specific infection or trigger for such severe tissue injury, ARDS carries a high mortality (particularly in those managed with mechanical ventilation and its associated lung injury), and it is also common for those fortunate survivors to have significant residual fibrosis, often persisting for months or years, sometimes not resolving but also rarely progressive.

This begs the question of which aspects of the pathophysiology are normal responses to tissue injury, and which are “aberrant”.

In this study, all samples are from individuals who died (though not necessarily all specifically from COVID), and the nature of this imitation, as acknowledged by the authors, are inherent in a study of this nature, but do pose a constraint to the degree to which such findings might contribute to an understanding of prolonged post-COVID injury/inflammation/fibrosis

Similarly the degree to which any of the proposed mechanisms might contribute to symptoms collectively referred to as “long COVID” is even more problematic.

In addition, the assumption that any of their observations are unique to COVID-19 is based in large measure on their limited control population: 4 individuals who had apparent bacterial pneumonia, as well as a number of comorbidities, leading to their demise, 2 of which underwent invasive ventilation and 2 of which did not.

Importantly none of them had ARDS, and their exact cause of death is not specified. As such, one can only reasonably surmise that the findings presented represent observations concerning acute lung injury/ARDS in general (though in 3 of the 13 COVID cases ARDS was absent, including one in the acute group, one in the chronic, and one in the prolonged).

We are grateful for the reviewer’s appreciation of our analyses as detailed and sophisticated.

The reviewer’s critical comments made us aware of the fact that in the previous version of the manuscript we did not sufficiently acknowledge the limitations of our study. We added a

sentence on the limitations associated with the analysis of autopsy tissues to the discussion part of the revised version. To address the reviewer's concern about the term "long-COVID" in this context, we decided to refrain from using it. We additionally cite a recent review, where they use the more specific term "post-COVID interstitial lung disease" (PC-ILD), and the authors show that this develops independently of ARDS during the acute phase¹. There is no correlation between ARDS and any of the parameters relevant to our conclusions, as analyzed by Pearson correlation and shown in Supplementary Fig. 1C, so we decided to refrain from using the term ARDS.

Moreover, we now pointed out that our observations may not be uniquely associated to COVID-19 in the last paragraph on the discussion. In the first version of the manuscript we did not sufficiently emphasize that it was never our aim to define mechanisms that are exclusively specific to COVID-19. On the contrary, we believe that -especially in view of the parallels to other chronic diseases associated with fibrosis, e.g. certain autoimmune diseases- the mechanisms we defined here could be interesting in a broader context, even beyond COVID-19. We are convinced that our study sets the technological basis to further explore those questions in prospective studies in the future.

These considerations aside, there are major concerns with the data which limit the degree to which the interpretations are convincing:

A) An important concern which permeates this manuscript is the manner of data presentation; it would behoove the authors to consider how they might present the data in such a way as to make the points they would like to make more clearly and persuasively, which would enhance the ability of readers to more readily judge the validity of their findings and interpretations. For example in Fig 1 multiple sections are shown from control specimens, as well as acute infection (14 days), chronic (34 days) and prolonged (51 days). H&E stains depict widespread abnormalities at low power, and a small "field of view" (FOV) has been selected as potentially informative (and presumably representative) by blinded pathologists.

Following the suggestion of the reviewer, we have taken special care to improve the data presentation and depict the complex data in a clearly understandable way in the revised manuscript. We have improved the resolution of all histological images for better interpretation. As mentioned in the text (Results part for Fig. 1, first sentence) and Figure legend (for Fig.1), the H&E, confocal and ST specimens represent consecutive sections of the same tissue block. Regarding the smaller field of view that was analyzed by MELC, we are indeed technically limited in the area we can analyze by highly multiplexed microscopy. However, to compensate for this, multiple FOVs of different, representative lung areas or tissue blocks of the individuals were analyzed. To transparently depict the differences between those FOVs, we are showing every single FOV as a data point in the respective graphs (See Fig. 1D, 2B, 4B and 5D)

Though the specific patients whose samples are shown are not identified (which would have been helpful) one can clearly see (not surprisingly) that intensity of collagen staining correlates with duration of illness.

The specific patients can be identified based on disease duration, which is also mentioned in the clinical data table. However, the reviewer's comment made us aware that this is

cumbersome for the reader, so we have added the information on the individual patients also to the Figure 1 of the revised manuscript.

There are other observations in the other panels, which might have been easier to appreciate with some carefully placed arrows, since depiction of the “juxtaposition” of different colors (which largely underpins their interpretations) is not always as obvious as the authors seem to believe.

We chose to present the immunofluorescence histology data in the form of overlays, as this is generally considered a standard for data presentation for this technique. Our intention was to change the histological images as little as possible and avoid covering of parts of the images for annotations. However, we agree that arrows placed with care could facilitate the interpretation of the data for a wide readership, therefore we have followed the suggestion of the reviewer and added arrows to Fig.1A to point out the immune cell accumulations. Concerning the increase in extracellular matrix and fibroblasts in prolonged cases, we also moved the supporting light sheet data (former Fig. 2G) to Figure 1, where only one single marker (ER-TR7, yellow) is depicted.

A critical issue is trying to understand the degree to which there is quantitative (or semi-quantitative) support for what are purported to be representative examples of their findings (though the interpretations are variably convincing by staring at which colors are next to which other colors (usually without a corresponding brightfield or other landmarks). Much must be taken on faith, particularly when staining patterns are marginally convincing.

We would like to emphasize that our whole approach on multiplexed histology analysis generates single-cell quantitative data. Therefore, we have previously established and published a whole image analysis pipeline², which we apply here, in combination with other spatial methods, to quantitatively assess the phenotype and localization of immune cells within tissues.

Regarding the concern of the reviewer for the lack of corresponding brightfield or landmarks, we want to emphasize that the information is shown in the manuscript in Supplementary Fig 2A. We have now added a reference to this figure in the text (see page 4).

B) In figure 2A for example, images from one case among the 4 groups show staining for smooth muscle actin, CD31, pancytkeratin, and ER-PR 7 (fibroblast marker). One has to surmise based upon the comments in the paper (and the legend) that the area at the bottom image from the prolonged case (which appears to show fibroblast staining interposed with faint CD31 staining) must be what the authors are calling a "fibrovascular niche", although the smooth muscle actin does not appear to overlap the fibroblastic staining.

As detailed in the text (see page 5), Fig. 2A illustrates (1) the absence/disruption of CD31 protein in the acute group (5 of 8 FOVs), (2) the absence/disruption of PCK in the acute group and (3) the increase in fibroblasts of different types in a representative prolonged case: myofibroblasts (SMA+, known to line intermediate-to-large vessels and airways to confer contractility) and reticular fibroblasts (ER-TR7+, which do not necessary reside around endothelial structures). The area the reviewer mentions (bottom image) was not meant to represent a fibrovascular/adventitial niche. Examples of adventitial niches are shown in Fig. 3 and 4, corresponding to the emergence of this concept in the text.

Fig 2B shows the expression levels of a number of endothelial genes PECAM 1, claudin 5, and von Willebrand's factor, in heat map fashion, of "average expression in each disease group" by spatial transcriptomics. If the authors more clearly delineated how representative these findings are, it might be more convincing that these do not simply reflect selectively presented data as well as what is to be concluded from this.

In this heat map, we show the average expression of the indicated genes analyzed, split by disease group. This includes the data from all ST sections, with no data left out, as indicated in the figure legend. In the revised version of the manuscript, we have extended our analysis and now included additional genes linked to endothelial integrity, which further support our hypothesis.

The point seems to be that endothelial markers are most down regulated in the acute case compared to the others (and though the single panel shown from the acute case seems to demonstrate complete absence of CD31 (in a single FOV), the text indicates that 5 out of 8 FOV's had no apparent CD31 staining (the panel shown has no cytokeratin staining either, and it seems vanishingly that both of the images are truly representative, since the case being shown (acute case 3) reported had no ARDS, and it is stated the COVID-19 was not part of the main cause of death.

Regarding endothelial damage, the reviewer's interpretation is in line with ours. Endothelial markers are downregulated in the acute group at the transcriptional level as measured by ST. Furthermore, CD31 expression at the protein level is absent in 5 out of 8 FOVs analyzed by multiplexed histology, similar to the loss observed in the epithelial marker PCK.

Regarding the reviewer's point on the cause of death, we understand this concern, and it made us realize that our labeling was not specific enough, we therefore listed now the cause of death after autopsy in analogy to the WHO guidelines³. The immediate cause of death is labeled with Ia, in addition to conditions leading to cause of death (Ib), underlying cause (Ic), and further relevant conditions (II). For the case mentioned by the reviewer, COVID-19 is listed in the autopsy report as Ib, so a condition leading to the cause of death. COVID-19 was the reason why the patient was hospitalized initially. This is also reflected in the high viral amounts measured in the patient's tissue. As highlighted in a recent review¹, post-COVID interstitial lung disease (PC-ILD) may occur without prior ARDS. Although our study is limited to autopsies, our data strengthen this concept, since -as pointed out by the reviewer- the case shown had no ARDS diagnosed.

To make our interpretation transparent to the reviewer, we are sending all of the 8 FOVs from acute cases including stainings for CD31 (cyan), SMA (blue), PCK (magenta) and ER-TR7 (yellow) along with this point-to-point response to the reviewer:

Acute case 1

Acute case 1

Acute case 2

Acute case 2

Acute case 2

Acute case 3

Acute case 3

Acute case 3

Figure 2D is another image of this case stained only for the fibroblast marker, shown only against autofluorescence, and the legend indicates that this is a large vessel with a thrombus attached to the vessel wall. This is not at all obvious, nor is the basis for making this determination. The supplemental video, while quite stunning, is no more informative. The interpretation appears little more than a guess, and one which is minimally supportive of their basic hypothesis at that.

While we agree with the reviewer that this is not a key result, we still decided to include the data as it supports the other results which show vascular dysfunction as a key

pathomechanism we describe in the paper. In the revised version of the Figure, as well as in the Video, we improved the annotation to point out the localization of the vessel and position of the thrombus. Autofluorescence was included to delineate the latter, as it is known to be highly abundant within those structures^{4,5}. We hope that this helps to interpret the image.

Also puzzling is the notation regarding Fig 2F, stating that "the image shows collagen 1 and CD45 both in white to depict tissue structure". Where is this white staining? The vessels are (manually) outlined in yellow, the fibrosis in green, and the airways in blue. Using spatial transcriptomics, genes associated with "blood vessel remodeling" are shown to be distributed differently in the acute case compared to the prolonged case, though the region which appears to have the most significant aggregation of local expression are in 3 foci in the field, only one of which is around a vessel. The largest aggregate concentration appears off to the bottom right of the image from the prolonged case, where there is no evidence of a vessel. One would like to see more robust support for what appears to be authors' "gestalt", which is not very convincing. Is there more rigorous evidence for this (as the focus of the manuscript)?

We agree that the white staining was difficult to see and apologize for that. This comment made us realize that our data presentation was suboptimal. We therefore have completely changed the presentation of those images. In addition, the comments of the reviewer made us aware that the previous data version of the figures mixed data on vascular dysfunction and the emergence of the adventitial niches. In the revised version, we decided to split the data and dedicate Figure 2 entirely to the vascular damage, while opening the concepts of adventitial niches in Figure 3. Along with that, we decided to move those data from former Fig. 2F to Figure 3. To facilitate comprehension of the spatial data, we now projected the annotations for vessels and bronchi on top of the ST data and changed the ST color code from rainbow to blue (Fig. 3C). This makes the identification of the various aspects of the "gestalt" easier to comprehend. Furthermore, we included the other two prolonged cases to Supplementary Fig 3D, in order to show all data and underline the concept.

Unsupervised clustering of gene expression from single cell sequencing is stated to demonstrate that cluster 5 and cluster 9 are co-located. Though cluster 5 appears most prominent around the vessel in the field, though cluster 9 is most impressively aggregated at the bottom right, near no vessel (cluster 9 is stated to include genes associated with "endothelial identity", which is puzzling. It is stated that cluster 5 and cluster 9 overlap with fibrotic landmarks (?) with the perivascular markers nearby, and this is simply not convincing. It is worth noting that Fig S4 apparently shows the highest expressing genes by cluster, but is completely unreadable, even with enlargement. Would it not be more reasonable to show 12 different heat maps (or spreadsheets) with the top 25 genes, in readable format?

We thank the reviewer for making us aware of the fact that we needed to be more precise in the presentation of the cluster analyses. We would like to emphasize that these data are not derived from single cell sequencing, but represent bulk sequencing with spatial resolution (spatial transcriptomics), which we also indicated in the results and the figure legends.

In order to better resolve each lung microenvironment by spatial transcriptomics analysis, we excluded the unresolved cluster that appeared in the later version of this manuscript and we re-clustered the data to increase the specificity and the resolution of the clusters. We also show in Fig 3B some relevant upregulated genes from the top 25 DEGs for each cluster to facilitate cluster (lung microenvironment) identification and cluster localization for the reader.

Along that, we have entirely rewritten the text related to this figure. We hope it is now easy and clear to follow.

Following the suggestion of the reviewer, we have now included the spreadsheets of the top 25 DEGs as raw data files. We still decided to leave the heat map in the Suppl. Fig. 3, albeit with increased resolution, to make it better readable. The addition of Fig. 3B, which includes relevant genes for each cluster will hopefully make it easier to relate cluster identity. All data are available at the GEO database, in accordance with the journal guidelines.

C) Figure 3 purports to show that fibrotic lung areas are enriched in CD163 and CCL 18, demonstrated in chronic case 5 and prolonged case 5, and neither of these interpretations are convincing based upon data shown. Based on the expression level scale to the right, both genes appear to actually be most downregulated around the presumptive fibrovascular niche in the prolonged case. CCL21 expression is shown to be up regulated in the region in question, and that seems convincing enough in the prolonged case 5.

In the new Fig. 3C, we placed the graph showing tissue localization of cluster 1 (enriched in CCL18 and CD163 transcripts, as shown in Fig. 3B) and cluster 2 (enriched in fibrosis-related transcripts) directly above the graph depicting GSEA of collagen biosynthesis pathways. It is evident that cluster 1 and 2 are largely adjacent, and together they overlap the area with increased gene expression related to collagen biosynthesis. CCL21 expression is indeed upregulated within adventitial niches as the reviewer mentions, overlapping with lung areas enriched in blood-vessel remodeling pathway. We hope all our points and the figures to sustain them are now convincing enough.

In figure B fibroblast proliferation appears to center around the structure in question as is to a lesser extent expression of collagen biosynthesis genes and EMT (stated to be "epithelial–mesenchymal" transition in the Legend, though the text is focused on endothelial–mesenchymal transition).

The reviewer brings up an interesting point here. The pathways involved in endothelial to mesenchymal transition and epithelial to mesenchymal transition are very similar, both involving trans-differentiation processes towards a mesenchymal phenotype and both promoted by overlapping signals⁶. We noticed that these pathways were enriched in areas landmarked as fibrovascular niches, therefore we assume that mainly EndMT is involved. However, we cannot rule out a contribution of epithelial to mesenchymal transition (EMT). To account for that, we have decided to use the term "adventitial" throughout the manuscript instead of "fibrovascular". We also adjusted the text as follows: "Unexpectedly, the microanatomical lung areas enriched in the epithelial to mesenchymal transition (EMT) pathway did not overlap with the epithelial C0 and C6, but rather with the adventitial C5 and C8. This suggests that tissue fibrosis not only arises from epithelial remodeling, but from the activated adventitial niches with a CCL21 signature. Thus, it likely represents endothelial to mesenchymal transition (EndMT)."

Figure 3C focuses on area of interest from a chronic case seemingly showing smooth muscle actin, CCR8, CCR7, ER–TR 7, and CD31. It's not clear what is being emphasized or offered as an example, but to the naked eye little of interest seems apparent.

Since we observed a specific chemokine signature in the ST data, our motivation was to check for the corresponding chemokine receptors and identify the cell types expressing those. Since

we observe both CCR7 and CCR8 receptors expressed within vessel walls that show signs of endothelial-to-mesenchymal transition (co-expression of endothelial and mesenchymal markers by the same cell), we conclude that the CCL21 and CCL18 can influence this process via their cognate receptors. We have added arrows to the figure and enlarged it to make our points more clear.

D) Figure 4 shows a field from a prolonged case #4, stated to represent a FVN, in which the large panel depicts staining for smooth muscle actin, CD3, CD31, and CCR7, in the region focused upon for the enlarged subpanels shows only the patch of CD3 staining. The panels to the right are stated to show the 4 stains individually, though the larger panel clearly shows much higher expression of CD31 and SMA outside of the selected region.

We changed the display in order to clearly mark the localization of the vessel and the immune cells in the surrounding adventitial niche. Based on the CD31 staining, we added the vessel outline in magenta, SMA staining is also present in those areas. There are certainly SMA+ and / or CD31+ structures in the tissue section a part from the structured outlined and referred to. Our claim is not that these markers are exclusively present in the activated FVNs that recruit T cells. Our claim is that all T aggregates in prolonged COVID-19 lungs are indeed found around activated adventitial niches (intermediate-to-large vessel structures) and that, thus, FVN (which we showed in Figure 3 are marked by CCL21 expression) host such accumulations of CCR7+ T cells with a particular phenotype. We have also added more T cell markers to the Figure 3A, to emphasize the importance of the broad exhaustion phenotype that the T cells localized there present, with some of them being proliferative T cells (Ki67+), resembling a progenitor population of exhausted T cells that has already been described in the context of chronic infections and cancer⁷⁻⁹.

The figure legend is entitled "Activated fibroblastic niches of prolonged COVID-19 lungs host CCR7 positive T-cell aggregates and imprint them with an exhausted, T follicular helper-like phenotype", though most of the data shown are expression data of CD4 cells and CD8 cells in lungs versus lymph nodes. Only figure 4G shows specific expression of CD8 and CD4 cell expression levels in "fibrovascular" (defined as in 4A?) versus epithelial regions, though it does not appear that any conclusion can be drawn from these heat maps, at least as presented.

In this figure, we demonstrate that T cells with an exhausted phenotype accumulate in the FVN. The draining lymph nodes of the same donors are shown to demonstrate that this phenotype is specific to the lung tissue, suggesting a local imprinting of the cells. The heat maps show that the phenotype of FVN T cells differs from the ones in epithelial areas, further underlining the role of the local microenvironment. We have worked on the text to make our point more clear. We also worked on representing the adventitial niche in a more clear way and added more immunofluorescence images in Fig 4, in order to emphasize the T cell phenotypes localized in those areas, and we changed the figure caption to account for this.

Neither is it evident (or stated) that statistical analysis has been applied to any of Fig 4 other than 4B.

We thank the reviewer for making us aware of this. The revised version includes the statistical analysis.

E) Fig 5 is a bit confusing since there are 2 figures labelled A and B, followed by 3 figures subsequently labeled A, B, and C.

This was a labeling mistake of the figure, for which we apologize. We have corrected that.

The legend for Fig 5 is entitled "CCL21+ fibrovascular niches aim at the formation of tertiary lymphoid structures in prolonged COVID-19 lungs". Sections shown are from prolonged case 5, a case noted to have been complicated by bacterial pneumonia (as noted in 2 of the 5 prolonged, and 4 of the 5 chronic cases).

The reviewer raises an important point, which motivated us to perform a correlation analysis between clinical metadata (shown in Supplementary Fig. 1C and 5D). This analysis showed that the formation of the observed lymphoid aggregates did not correlate with bacterial pneumonia, but instead with the fibrosis score and sample stratification.

We have added the data from two more prolonged cases, one of those without diagnosed bacterial pneumonia, and changed the text as follows: "Similar large, dense and structured immune cell aggregates, showing such unique transcriptional fingerprint reminiscent of iBALT could be observed in 2 out of 3 prolonged lungs analyzed by ST (Supplementary Fig. 5B, C). In addition to that, smaller lymphoid aggregates containing both Pax5+ B cells and CD4+ T cells, and associated to PNAd+ high endothelial venules, were found in another prolonged case (Fig. 5C)."

In Fig 5A(#1) CD163 and CCL 18 expression by lung macrophages is shown, and there is significant divergence in CCL18 in chronic and prolonged (it would be helpful to know whether these were from 2 regions of interest, different patients, or what exactly is being shown.

As stated above, there was mislabeling of the figures, which must have led to confusion. We apologize for this avoidable mistake. The figure mentioned by the reviewer is now Supplementary Fig. 3F. It shows snRNAseq data. We aimed to include this to support our data on CD163 and CCL18 colocalization. These data actually demonstrate that CD163+ macrophages are indeed the cellular source of CCL18 in COVID-19 lungs.

In Fig 5A(#2) a region of interest is outlined which is an airway stated to be "a highly infiltrated bronchus" with a dense immune cell aggregate adjacent to it in the left lower corner. The bronchus presumably contains a mucous plug (possibly with CD45+ PMN?; the patient is listed as having had secondary bacterial pneumonia). And while the CD45+ cell aggregate near it may represent bronchus-associated lymphoid tissue, supported by the expression of CD3, PAX5, lymphotoxin-beta, CXCL13), it cannot be determined to be a "tertiary lymphoid structure" without architectural information. It is clearly a small lymphoid aggregate, based upon gene expression, it would seem entirely possible that the aggregate is forming in response to the secondary infection. Surely there are other examples of "true tertiary lymphoid structures" if they are real (and there were 3 prolonged cases without secondary bacterial pneumonia). One might speculate that tertiary lymphoid structure may be in the process of forming in this panel, ("aiming"?) but surely some must have formed somewhere in one of these lung specimens.

In order to provide more architectural information we have performed additional histological analyses. We show a case without secondary bacterial pneumonia, displaying a lymphoid aggregate, characterized by the accumulation of T cells, B cells and even a PNAd+ structure with a lumen, suggestive of a high endothelial venule (HEV), in Figure 5. Despite the presence of HEV and clearly visible contacts occurring between CD4 T cells and B cells, indicating an

organized structure, we decided to refrain from naming this a tertiary lymphoid structure and rather use the term iBALT in the text.

In addition, we added data from another case, further supporting our concept, to Suppl. Fig. S5.

F) Fig S6A shows multiple panels each with multiple overlapping stains, and it is quite difficult to visualize the overlaps that are interpreted to reflect "non-conventional" T cells. In the bottom right subpanel of prolonged case 4, it would appear that ICOS and CD161 are co-localized, but the total number of stained cells (at the resolution in the figure) is not enough to be convincingly interpretable. Do the authors have better, or more extensive, representative examples? Would DAPI staining help delineate the cells in question? Are there other data to support the conclusion?

The purpose of this supplementary figure (Supplementary Fig. 4 in the revised version) is to give the reader a qualitative example of the exhausted T cell aggregates around adventitial niches in prolonged lungs (additionally expressing variable levels of several functional /activation markers) , in addition of the dot plots and graphs summarizing the marker expression in Fig. 4C, D, E and G. Quantification of the multiplexed immunofluorescence data was performed on all MELC sections using the pipeline for multiplexed image analysis², where 17.303 T cells were fed into the clustering analysis (12.534 from the LN and 4.769 from the lungs). Our segmentation pipeline is suited to robustly identify hematopoietic and structural cells within tissues, as previously demonstrated². It actually uses a combination of positivity for the nuclear stain DAPI, and a sum image for membrane/cytoplasmic markers to delineate single cells, thereby using an approach similar to the one suggested by the reviewer.

Regarding the term “non-conventional T cells”, we decided to refrain from using it and instead to use the more precise terminology “exhausted T cells”, in order to focus the discussion.

We have added another prolonged case with the respective markers in Suppl. Fig. 4 to make the phenotyping more representative. In addition, we include here in this response additional images for aggregates of exhausted T cells within lungs from prolonged cases:

G) The discussion is littered with statements that reflect considerable overinterpretation of morphometric data, but also a curious conflation of post-Covid lung injury/fibrosis with "long-Covid", and there is little to support a relationship between these descriptive entities. The text

refers to co-expression of PD1 and ICOS, which would not be surprising, but there are no such data apparent.

Our intention was to point out that some of the pathomechanisms described in our work might explain the long-term lung dysfunction and fibrosis, even after clearance of the virus. However, we agree that there is little to support a relationship between the data generated from post-mortem tissue analysis and Long-COVID. That is why we took out the emphasis for this potential, but poorly supported link, from the text and we rather link our observations to newly emerged and better mirrored disease entity: “post-COVID interstitial lung disease”¹.

As we focused our discussion on T cell exhaustion, we took out the sentence on PD-1 and ICOS co-expression.

We would like to point out, however, that the data in the manuscript are not plain morphometric data, but rather a comprehensive phenotyping of cells within tissues, in combination with spatial information at the protein and at the transcriptional level.

The "data strongly suggesting the formation of iBALT in prolonged Covid-19 lungs"... "linking tertiary lymphoid structures to lung pathology and fibrosis" is hardly justified by the data.

We have toned down the conclusion on the formation of tertiary lymphoid structures (actually we are not using this term anymore) and have taken the sentence out.

References:

1. Mehta, P., Rosas, I. O. & Singer, M. Understanding post-COVID-19 interstitial lung disease (ILD): a new fibroinflammatory disease entity. *Intensive Care Med.* (2022) doi:10.1007/s00134-022-06877-w.
2. Pascual-Reguant, A. *et al.* Multiplexed histology analyses for the phenotypic and spatial characterization of human innate lymphoid cells. *Nat. Commun.* (2021) doi:10.1038/s41467-021-21994-8.
3. WHO. MEDICAL certification of cause of death; instructions for physicians on use of international form of medical certificate of cause of death. *The Journal of the Egyptian Medical Association* vol. 35 287–288 at (1979).
4. Shrirao, A. B. *et al.* Autofluorescence of blood and its application in biomedical and clinical research. *Biotechnology and Bioengineering* at <https://doi.org/10.1002/bit.27933> (2021).
5. Flaumenhaft, R. *et al.* Localization and quantification of platelet-rich thrombi in large blood vessels with near-infrared fluorescence imaging. *Circulation* (2007) doi:10.1161/CIRCULATIONAHA.106.643908.
6. Saito, A. EMT and EndMT: regulated in similar ways? *Journal of Biochemistry* at <https://doi.org/10.1093/jb/mvt032> (2013).
7. Dähling, S. *et al.* Type 1 conventional dendritic cells maintain and guide the differentiation of precursors of exhausted T cells in distinct cellular niches. *Immunity* 55, (2022).

8. Vannella, K. M. *et al.* Transforming growth factor- β -regulated mTOR activity preserves cellular metabolism to maintain long-term T cell responses in chronic infection. *Immunity* **54**, 1–14 (2021).
9. Utzschneider, D. T. *et al.* Early precursor T cells establish and propagate T cell exhaustion in chronic infection. *Nat. Immunol.* (2020) doi:10.1038/s41590-020-0760-z.

Reviewer #2 (Remarks to the Author):

This manuscript is important and contains extremely valuable information from an interrogation of lung tissues by multiple methods resulting in a compendium of information. This is now much improved and all of the previous concerns of this reviewer have been thoroughly addressed.

Reviewer #3 (Remarks to the Author):

I would like to thank the authors for their consideration of my assessment and criticism. My questions have been sufficiently addressed. In my view, there are no further revisions required.

Reviewer #4 (Remarks to the Author):

This manuscript is significantly improved, and makes it somewhat easier to understand the degree to which the observations described are convincing. The mechanistic hypotheses (not conclusions) still reflect attempts to make the data fit a story, and for every seemingly linear cause/effect relationship espoused on the basis of static expression data, there are likely a multitude of possible mechanistic hypotheses which represent alternative (equally reasonable) interpretations.

Point to point response: NCOMMS-22-09465B

Reviewer #2 (Remarks to the Author):

This manuscript is important and contains extremely valuable information from an interrogation of lung tissues by multiple methods resulting in a compendium of information. This is now much improved and all of the previous concerns of this reviewer have been thoroughly addressed.

We thank the reviewer for the positive evaluation and for highlighting the importance of this work.

Reviewer #3 (Remarks to the Author):

I would like to thank the authors for their consideration of my assessment and criticism. My questions have been sufficiently addressed. In my view, there are no further revisions required.

We thank the reviewer for the appreciation and positive evaluation of the revised version of the manuscript.

Reviewer #4 (Remarks to the Author):

This manuscript is significantly improved, and makes it somewhat easier to understand the degree to which the observations described are convincing. The mechanistic hypotheses (not conclusions) still reflect attempts to make the data fit a story, and for every seemingly linear cause/effect relationship espoused on the basis of static expression data, there are likely a multitude of possible mechanistic hypotheses which represent alternative (equally reasonable) interpretations.

We are glad that the reviewer considers the revised manuscript improved and easier to understand. To emphasize that our data do allow other possible mechanistic hypotheses, we partly adopted the wording of reviewer 4 and added a sentence emphasizing the possibility of alternative interpretations to the discussion. We also toned down our conclusions throughout the text.